# Hierarchical Planning for Complex Tasks with Knowledge Graph-RAG and Symbolic Verification

**Flavio Petruzzellis** [1]   **Cristina Cornelio** [2]   **Pietro Liò** [3]

## Abstract

Large Language Models (LLMs) have shown promise as robotic planners but often struggle with long-horizon and complex tasks, especially in specialized environments requiring external knowledge. While hierarchical planning and Retrieval-Augmented Generation (RAG) address some of these challenges, they remain insufficient on their own and a deeper integration is required for achieving more reliable systems. To this end, we propose a neuro-symbolic approach that enhances LLMs-based planners with Knowledge Graph-based RAG for hierarchical plan generation. This method decomposes complex tasks into manageable subtasks, further expanded into executable atomic action sequences. To ensure formal correctness and proper decomposition, we integrate a Symbolic Validator, which also functions as a failure detector by aligning expected and observed world states. Our evaluation against baseline methods demonstrates the consistent significant advantages of integrating hierarchical planning, symbolic verification, and RAG across tasks of varying complexity and different LLMs. Additionally, our experimental setup and novel metrics not only validate our approach for complex planning but also serve as a tool for assessing LLMs' reasoning and compositional capabilities. Code available at https://github.com/corneliocristina/HVR.

## 1. Introduction

Humans naturally tackle complex tasks, such as boiling water, by first conceptualizing high-level steps (e.g., filling a pot with water) and then breaking these down into a series of individual actions (e.g., picking up the pot, carrying it to the sink, and turning on the faucet). This hierarchical process allows humans not only to plan the sequence of actions but also to ensure the plan is feasible before starting. Additionally, during execution, humans constantly monitor the task, identifying and addressing any issues that may arise (e.g., realizing the pot is dirty or the faucet isn't working). For agents to replicate this capability, they need to manage actions at varying levels of abstraction, know how to interact with objects in their environment, and detect and respond to planning or execution issues. These characteristics are particularly crucial for complex or long-horizon tasks.

**LLMs as planners.** Recently, Large Language Models (LLMs) have emerged as promising tools for planning tasks in agentic systems (Huang et al., 2022). Their ability to process natural language (NL) and generate structured outputs makes them appealing for robotic task planning. Despite this potential, they continue to struggle with long-horizon tasks due to a lack of hierarchical reasoning and insufficient knowledge of the specific properties of objects and environments. These limitations have driven the development of hierarchical planning, which decomposes complex tasks into multiple abstraction levels (Höller et al., 2020). While this approach improves long-horizon robotic planning, it still faces challenges in robustness and adaptability in complex environments. To address knowledge limitations, Retrieval-Augmented Generation (RAG) methods have been introduced, enabling systems to retrieve task-relevant information from external sources (Lewis et al., 2020). Such approaches are particularly crucial in specialized settings, like healthcare (e.g., surgical robots), automated transportation (e.g., spacecraft, autonomous vehicles), and domestic assistance (e.g., kitchen robots), where general commonsense knowledge falls short. However, while RAG improves access to relevant information, LLM-generated plans still lack the precision and reliability required for real-world robotics (Xie, 2020). LLMs generate plausible action sequences relying on statistical correlations rather than logical inference, leading to inconsistencies and errors. To mitigate these issues, symbolic verification is essential for ensuring correctness before execution, reducing the risk of failures. Some methods attempt to translate NL instructions into for-

[1]Department of Mathematics, University of Padova, Padova, Italy [2]Samsung AI, Cambridge, UK [3]Computer Science Department, University of Cambridge, Cambridge, UK. Correspondence to: Cristina Cornelio <c.cornelio@samsung.com>.

*Proceedings of the 42nd International Conference on Machine Learning*, Vancouver, Canada. PMLR 267, 2025. Copyright 2025 by the author(s).

mal languages (Liu et al., 2023a) or integrate verifiers into planning pipelines (Xie, 2020), but they still face challenges in ensuring scalability and adapting to the complexities of real-world environments. Thus, while LLMs offer promising capabilities for planning, their deep and effective integration with hierarchical reasoning, retrieval, and symbolic verification remains an open challenge.

In this work, we introduce **HVR**, a method that enhances LLM planning capabilities through a novel neuro-symbolic integration of **H**ierarchical planning, symbolic **V**erification and reasoning, and **R**AG methods over symbolic Knowledge Graphs. By decomposing complex tasks into manageable sub-tasks, our approach simplifies the planning process, while retrieving external information about the environment improves accuracy and reduces hallucinations. A Symbolic Validator further enhances reliability by simulating the plan in an "ideal world", enabling the identification and correction of potential issues before execution. Additionally, it acts as a failure detector, monitoring discrepancies between the ideal and real-world states during execution, allowing the system to adapt and correct itself in real time (Cornelio & Diab, 2024; Liu et al., 2023b).

**Bridging Model-Based & Policy-Based Approaches.** Traditionally, autonomous robotics has relied on two main approaches (Chatzilygeroudis et al., 2019): model-based systems, enabling an agent to simulate future states and rewards using an internal model of the environment; and policy-based systems, focusing on learning optimal action strategy based on the current state and the goal of the task. We propose a hybrid method where an ontology is used both to model the environment and to refine an LLM-generated policy along with a Symbolic Validator, enhancing its adaptability and robustness, particularly in situations with limited data and out-of-distribution cases. An additional advantage of our method is the ability to create a reusable macro-action library, allowing agents to share and reuse plans, reducing the need to regenerate plans. These macro actions can be shared, further organized, and refined by clustering similar actions from different robots, extracting high-level commonalities across varied hardware and capabilities.

**Major contributions and results summary.** The major contribution of this work is HVR, a novel neuro-symbolic system that integrates high-level reasoning with hierarchical planning, efficient information retrieval with RAG over a symbolic knowledge graph, and formal verification to enhance planning accuracy and formal correctness. Additionally, we introduce a diverse set of metrics and tasks designed for robotic simulators. These not only evaluate the effectiveness of planning systems like HVR, but also provide a comprehensive framework for assessing the reasoning and compositional capabilities of LLMs. Importantly, we selected freely available LLMs to ensure reproducibility.

While newer, paid models may offer improved general performance, they would also benefit from our method, particularly in complex tasks that require specialized knowledge.

Our results demonstrate that HVR significantly outperforms all baselines across both small and large LLMs, as well as tasks spanning different levels of complexity, from moderate to high. For smaller LLMs, RAG plays a crucial role in compensating for their limited reasoning capabilities, while for larger LLMs, hierarchical planning emerges as the most impactful feature. The combination of these two approaches, coupled with symbolic verification, achieves the best overall performance. Our analysis reveals that LLM-based planners still tend to generate unnecessarily long plans with extra steps, indicating areas for further improvement. Moreover, while performing very well on tasks with specific goal states, they struggle with more generic or open-ended objectives.

**State of the Art.** Recent research has explored the use of pre-trained LLMs for planning tasks in interactive environments, leveraging their embedded knowledge to generate plans (Huang et al., 2022; Wu et al., 2023; Ahn et al., 2022; Yao et al., 2020), but these methods often face hallucination and lack robust reasoning (Golovneva et al., 2022; Ahn et al., 2022). Hierarchical planning has been studied in combination with LLMs, such as HiP for symbolic planning combined with visual grounding (Ajay et al., 2024), and MLDT for multi-level decomposition (Wu et al., 2024). RAG systems (Lewis et al., 2020), which retrieve task-specific knowledge from sources like text (Guu et al., 2020), knowledge graphs (Edge et al., 2024), or the web (Kim et al., 2024), have also proven effective, especially for long-horizon tasks (Lu et al., 2023). Recent works employ external verifiers to refine LLM planning, as seen in the LLM-Modulo framework (Kambhampati et al.), where tools like CLAIRIFY (Skreta et al., 2023) ensure syntactic correctness, and symbolic verifiers improve LLM performance by validating plans (Valmeekam et al., 2023). Our work takes inspiration from these methods, integrating symbolic verification and hierarchical planning within an RAG-enhanced LLM-based planner to address complex, long-horizon tasks. However, most tasks addressed by current state-of-the-art models are very simple, such as rearranging blocks or retrieving objects in an environment, and do not reach the complexity of even the simpler tasks in our evaluation pool. See Appendix A for a detailed discussion of state-of-the-art approaches and their differences from our work.

## 2. The HVR Method

The problem we address is generating a plan to make a robot execute a task in an environment $\mathcal{E}$, based on a description of the task in natural language (NL) and an ontology $\mathcal{O}$ describing $\mathcal{E}$. Figure 1 provides an overview of our method, while Figure 2 illustrates the agent's execution and feedback.

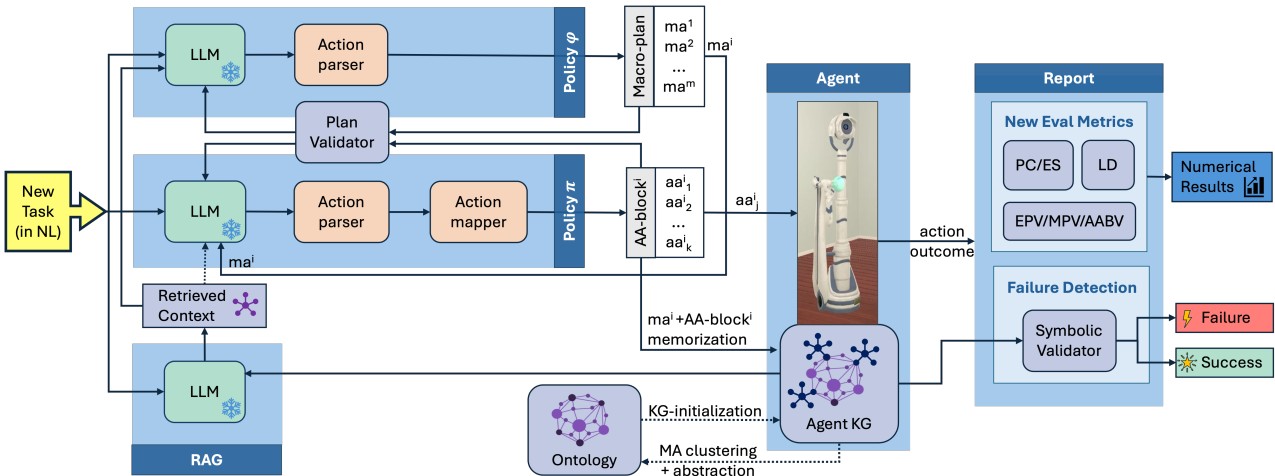

*Figure 1.* Overview of HRV: (1) Given a natural language task description, a pre-trained frozen LLM to generate a macro-plan (policy $\varphi$), which is expanded into an AA-block (policy $\pi$). Retrieval-augmented generation (RAG) method retrieves relevant context from the agent knowledge graph (initialized with the environment ontology) to support plan generation, while a Plan Validator detects and triggers the correction of potential errors. (2) Once the plan is finalized, the agent executes the atomic actions (AA) from each AA-block within the environment (see Figure 2), while recording the execution details in the knowledge graph. (3) After each action, a Symbolic Validator verifies the alignment between the "ideal plan" and the actual environment state, potentially detecting failures. The system then reports the performance of the LLM-based planner using a set of novel metrics.

## 2.1. Ontology & Dynamic Knowledge Graph

We adopted the approach in Cornelio & Diab, 2024, which utilizes symbolic knowledge to support planning. This knowledge is provided as an ontology $\mathcal{O}$ describing a kitchen environment, robot actions, and human preferences. The ontology includes a taxonomy of abstract object classes (uppercase, e.g., *Apple*) and defines relationships between them (e.g., *Apple* is a subclass of *Sliceable*). During task execution, the ontology is combined with real-time instances (lowercase, e.g., *apple-1*), representing concrete objects, agents or events. Together, the ontology $\mathcal{O}$ and these instances forms a Knowledge Graph $\mathcal{G}$, containing a set of relationships represented by either unary (indicating the class of an entity) or binary (relations between entities) predicates. The Knowledge Graph $\mathcal{G}$ is initialized as a copy of the ontology $\mathcal{O}$. As the robot executes actions, it records each new event instance in $\mathcal{G}$ along with the multi-modal action outcome. Visual data is converted into triples representing the scene-graph captured by the robot camera while auditory information is labeled and stored into triples.

## 2.2. Knowledge Graph RAG (R)

Retrieval-Augmented Generation (RAG) (Lewis et al., 2020; Edge et al., 2024) is a well-established technique used in tasks like question answering, planning, or image generation, especially when the task requires additional information from external sources (e.g., text or knowledge graphs) to enhance generation. In our work, we specifically focus on Graph RAG methods, utilizing Knowledge Graphs (KGs)

for retrieval, which are particularly suited for efficient querying and reasoning over symbolic data. A general KG-based RAG process starts with a query formulated based on the task requirements. This query is then used to search within the KG, retrieving nodes (entities) and edges (relationships) relevant to the task. The result is a subgraph composed of these nodes and edges, which serves as the context for the LLM, that can be either directly included in the LLM prompt or mapped to an embedding space that the model can understand. In our approach, we use an LLM as a planner, utilizing a KG-based RAG to enhance plan generation by exploiting both the retrieved and the LLM implicit knowledge. Specifically, we extract a subgraph $\mathcal{G}'$ (of the full KG $\mathcal{G}$), containing instances of the objects required for the plan, along with their corresponding types, properties, and state.

## 2.3. Hierarchical Plan generation (H)

The **input** to our planning problem consists of a NL text $t$ including (1) the task definition, and (2) a subgraph $\mathcal{G}'$ extracted from the knowledge graph $\mathcal{G}$, containing relevant information for achieving the goal. The task definition can be provided in either a *constructive form*, which outlines instructions on how to solve the task (e.g., "boil water in a pot and place it on a countertop"), or in a *goal-oriented form*, describing the desired end state (e.g., "I want a pot with boiling water on a countertop"). The final **output** of the system is a sequence of *atomic actions* (AAs), which can be directly executed by the robot (e.g., *put-on(pan, stove)*). We refer to the complete set of all possible atomic actions the robot can execute within the environment as the *action*

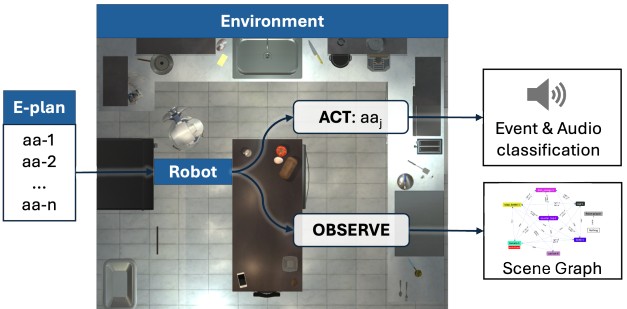

Figure 2. An agent executes an Expanded plan in a kitchen environment following its sequence of atomic actions $aa_j$. For each action, the robot interACTs with the environment and OBSERVEs the resulting state. Visual observations are captured as a scene graph, representing object relative positions and states, while auditory feedback is processed through classification. This multi-modal outcome corresponds to the Agent's *action outcome* in Fig. 1.

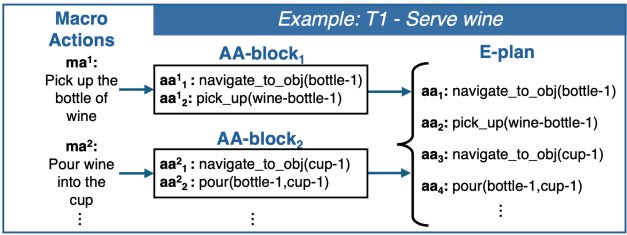

Figure 3. The task "Serve wine" is divided into multiple macro actions MAs $(ma^1, \ldots, ma^m)$ such as "Pick up the bottle of wine" and "Pour wine into the cup". Each macro action is decomposed into a AA-block with atomic actions AAs $(aa_1^i, \ldots, aa_{k_i}^i)$ such as *navigate_to_obj*, *pick_up*, and *pour*. The final expanded plan (E-plan) is the concatenation of all the AAs from each AA-block.

set $\mathcal{A}$, and this set is assumed to be finite.

Since for complex tasks the sequence of atomic actions (AAs) can become quite lengthy, we decompose the task into a series of subtasks called *macro actions* (MAs) $(ma^1, \ldots, ma^m)$. Each macro action $ma^i$ is a high-level, concise NL description of the subtask goal and can be further expanded into a sequence of AAs $(aa_1^i, \ldots, aa_{k_i}^i)$. We refer to each ordered sequence of $k_i$ AAs expanding a macro action $ma^i$, as an *AA-block*. These AA-blocks are then concatenated to form the final *expanded plan* (E-plan) $\langle aa_1, \ldots, aa_n \rangle$, where $n = \sum_{i=1}^{m} k_i$. See Figure 3 for an example. It is important to notice that RAG is employed at each step both for generating MAs and AA-blocks, as requirements for different MAs might vary and the KG is dynamic: object states change over time, and new objects can be created/destroyed (e.g., through actions like slicing).

The **generation of a Macro Plan (M-plan)** is performed using a task-conditioned policy $\varphi$, which is a function mapping a NL text $t$ to a sequence of macro actions (MAs):

$\varphi : t \mapsto P_M$, where $P_M = \langle ma^1, \cdots, ma^m \rangle$. In our approach, $\varphi$ corresponds to the combination of a pre-trained frozen LLM and an action parser: $\varphi(t) = parse(LLM(t))$. The LLM associates the input text $t$ with a transformed text $t'$ through the mapping $LLM : t \mapsto t'$. Subsequently, the action parser $parse(\cdot)$ processes the generated text $t'$ and extracts each line as an action string. The **generation of an AA-block** is achieved using a task-conditioned policy $\pi$, which is a function that maps a macro action $ma^i$ to a sequence $P$ of atomic actions: $\pi : ma^i \mapsto P$, where $P = \langle aa_1^i, aa_2^i, \ldots, aa_{k_i}^i \rangle$ with $aa_j^i \in \mathcal{A}$, and $k_i$ the length of the AA-block. Similar to the macro plan generation, $\pi$ also employs a pre-trained frozen LLM combined with an action parser. However, it introduces an additional component, an action mapper $\mu$: $\pi(t) = \mu(parse(LLM(t)))$ which associates the transformed input text $t'$ with a sequence of AAs, mapping it as $\mu : t' \mapsto P$. The action mapper identifies the corresponding action in the action set $\mathcal{A}$ that is closest in the embedding space to each action string (see Section 3). The policy $\pi$ can also be used to generate the full E-plan directly from the task description, bypassing the generation of MAs.

## 2.4. Plan Verification and Correction (V)

We represent both AAs and MAs using a syntax based on the Planning Domain Definition Language (PDDL) (McDermott et al., 1998), a formal language widely employed in automated planning systems that provides a standardized framework for defining objects, actions, and their associated preconditions and effects. In PDDL, a task is defined in two main parts: (1) the *domain*, which describes the atomic actions (e.g., *pick_up*, *toggle_on*) along with their respective pre and postconditions; and (2) the *problem*, which outlines the initial state of the environment and the target goal. In our approach, the atomic actions in the domain correspond to $\mathcal{A}$, and consists of agent-specific actions with explicitly defined pre- and postconditions, as the agent's available capabilities are known and finite. On the other hand, MAs can cover an unlimited range of combinations of actions in $\mathcal{A}$. Thus, it is impractical for domain experts to manually specify all the possible preconditions and effects that would determine the feasibility of each MA. To address this, we leverage the LLM to dynamically generate these conditions.

**Heuristic-based Plan Verification and Correction.** For each given macro plan, expanded plan, or AA-block, we initially employ a deterministic heuristic to adjust actions, such as inserting any missing navigation steps. This process leverages information defined in the PDDL. For instance, if the original sequence of actions is *(pick-up, tomato-1), (put-on, tomato-1, countertop-1)*, the heuristic would correct it to *(navigate-to-obj, tomato-1), (pick-up, tomato-1), (navigate-to-obj, countertop-1), (put-on, tomato-1, countertop-1)* to ensure the plan includes all necessary navigation steps.

**Symbolic Plan Verification.** For a given macro plan, expanded plan, or AA-block, we utilize a Symbolic Validator to assess its feasibility. The validator indicates whether the plan is valid or, if not, identifies the step where the error occurred specifying which particular condition was violated. VAL (Howey et al., 2004) is an example of a widely used verification tool for PDDL plans.

**LLM-based Plan Correction.** Our system employs back-prompting with LLMs to correct plans using feedback from the Symbolic Validator, addressing two types of errors: (1) When an error is identified in a macro plan, the LLM adjusts the pre and postconditions by adding, removing, or modifying them based on the feedback; (2) If the error is found in an expanded plan or AA-block, the LLM corrects the sequence of AAs by modifying, adding, or removing actions. For macro plan corrections (1), the LLM receives as input the task description, the plan generated by the policy $\varphi$ based on that description, the original list of pre- and post-conditions for each MA, and details on the violated condition. Additionally, the LLM is provided with a list of relevant objects retrieved with RAG with their properties (e.g., *pickupable*, *cookable*), states (e.g., *closed*, *open*, *picked_up*), and triples about object locations (e.g., *(object1,inside,object2)*), but no in-context examples of correction are supplied. The LLM then generates an updated list of pre- and post-conditions for each MA. For AA-block corrections (2), the LLM uses the error information from the Symbolic Validator, along with the MA description and the sequence of atomic actions executed up to that point. The model also receives one in-context example of correction and produces an updated version of the AA-block.

**Aligning observations and expected world state.** At runtime, we employ the same PDDL description to verify if the expected environment state aligns with the actual state. This is done by aligning symbolic information from the scene graph (encoding the visual data captured by the agent) with the expected state generated by executing the PDDL in a planner. The Symbolic Validator thus acts as a failure detector: if a discrepancy is found between the "ideal world" triples and those in the scene graph, the action's expected pre- or post-conditions were not met in the environment, suggesting incorrect execution. Upon identifying a failure, established correction techniques (Cornelio & Diab, 2024; Liu et al., 2023b) can be applied.

### 2.5. Macro Action Library and Knowledge Transfer

A key advantage of our method is the ability to build a reusable library of macro actions that can be accessed by the same agent for future tasks or shared across multiple agents. This eliminates the need to regenerate plans from scratch, facilitating knowledge transfer and enabling agents to benefit from a collective "culture" of successful actions and their

*Table 1.* Tasks implemented in the experiments.

| | ID | Name | steps | objects |
|---|---|---|---|---|
| moderate complexity | T1 | Serve wine | 8 | 2 |
| | T2 | Make coffee | 9 | 2 |
| | T3 | Fry egg in a pan | 13 | 2 |
| | T4 | Toast bread | 15 | 3 |
| | T5 | Warm water (in microwave) | 16 | 3 |
| | T5bis | Warm water (generic) | 16 | 3 |
| | T6 | Cook potato slice (in microwave) | 20 | 3 |
| high complexity | T7 | Salad | 26 | 4 |
| | T8 | Vegan sandwich | 30 | 5 |
| | T9 | Cook egg and potato slice | 32 | 7 |
| | T10 | Complex salad | 33 | 7 |
| | T11 | Tomato-egg on toast | 38 | 10 |
| | T12 | Complex plate | 41 | 10 |

corresponding atomic steps. During task execution, both AAs and MAs are recorded in the agent's Knowledge Graph $\mathcal{G}$. Once the task is successfully completed, the macro actions are then transferred from $\mathcal{G}$ to the ontology $\mathcal{O}$. These actions are then organized by clustering similar macro actions, even if they have been executed by different robots with varying hardware (e.g., single- or dual-arm robots) or capabilities (e.g., a robot that can pick up objects versus one that can only push objects). The clusters are refined using established techniques such as summarization or abstraction to identify high-level commonalities. This process is depicted in Figure 1 with dotted lines.

## 3. Experiments Setup

### 3.1. AI2Thor simulator and OnthoThor Ontology.

AI2Thor is a 3D simulator that provides a realistic virtual environment where agents can navigate and interact with objects in a home setting. Following previous studies (e.g., Cornelio & Diab, 2024; Liu et al., 2023b), we focused on the kitchen domain. We used AI2Thor as state transition function $\mathcal{T}(s, aa) = s'$, which maps an environment state $s$ to the next state $s'$ through the application of an atomic action $aa$. Formally, AI2Thor acts as $\mathcal{T} : \mathcal{S} \times \mathcal{A} \rightarrow \mathcal{S}$ where $\mathcal{S}$ denotes the state space of the environment $\mathcal{E}$.

We employed *OntoThor* (Cornelio & Diab, 2024), which describes the AI2Thor kitchen environment, as our ontology. This ontology includes not only classes representing agents, physical objects, and their properties but also the framework to handle events, actions, and to capture audio-visual inputs from the robot sensors in the form of triples.

### 3.2. Tasks

We chose to conduct our experiments on the tasks defined in (Cornelio & Diab, 2024) for their complexity, as most tasks tackled by state-of-the-art models are simple, such as rearranging blocks or retrieving objects, and do not match

the complexity required for our study. In addition, we introduced two new complex tasks, T11 and T12[1], and we duplicated T5 as T5bis: in T5, the robot must warm water specifically using the microwave, whereas in T5bis, the robot can warm the water using any available method, such as on a stove, in a kettle, or in the microwave. Table 1 provides an overview of these tasks, organized by the number of steps in the ground truth plan. The tasks are divided into two categories: **moderate complexity tasks** with up to 20 steps, and **high complexity tasks** with more than 20 steps and involving more than three objects. The ground truth plans are minimal, containing only the essential steps required for completion without unnecessary actions.

### 3.3. Implementation details.

**Knowledge Graph RAG.** To provide the LLM-planner with relevant, up-to-date information about the environment's state, we use a Knowledge Graph Retrieval-Augmented Generation (KG-RAG) approach. The same frozen LLM identifies a subset of relevant objects from those currently available in the kitchen, as stored in the KG. Using a zero-shot prompt, we provide the LLM with a natural language task description and a complete list of objects (a feasible step due to the manageable size of OntoThor). The LLM selects a subset of objects it deems necessary for the task, which are then mapped to their corresponding instances in the KG using cosine similarity over their embeddings. We then symbolically query the KG to retrieve relevant information about the selected objects, including their current state (e.g., *open*, *closed*, *cooked*), inherent properties (*receptacle*, *cookable*, *pickable*) and technical capabilities (e.g., appliances being *toggleable*).

**LLM acting as a planner.** To generate macro actions, we employ a two-shot prompt template that incorporates the goal-oriented task description and the list of relevant objects with their properties, retrieved by the KG-RAG approach. A frozen LLM is fed with this prompt and outputs an ordered list of macro actions in natural language. To generate AA-blocks for a given macro action (described in natural language), the LLM is first asked to generate a verbal description of the steps needed to complete the task, followed by a sequence of atomic actions for the robot to execute. The prompt briefly outlines key environment informations, such as the robot's capabilities in the AI2Thor simulator, its single-arm limitation, and how to interact with kitchen appliances, including one example for each. Additionally, the LLM is provided with the history of executed atomic actions, grouped by macro action if hierarchical planning is enabled. To ensure consistent results, we employ in-context learning, providing two examples of plans for simple macro

---

[1]Tasks were renumbered from (Cornelio & Diab, 2024); see Appendix B.4 for details on the mapping.

actions. Both the prompt template for generating macro actions and the one for generating AA-blocks include an updated list of objects in the environment along with their properties. When KG-RAG is enabled, this list is limited to only task-relevant objects.

In our experiments we implemented the LLM-based planner using both `Phi-3-mini-4k-instruct` (Abdin et al., 2024), a small-scale open-source LLM specialized on reasoning tasks, and `gemini-1.5-flash` (Anil et al., 2023), a larger closed-source model with stronger reasoning capabilities, also admitting a bigger context. We selected freely available LLMs to ensure reproducibility.

**Action parser and action mapper.** We use an action parser to extract both macro and atomic actions from the LLM output by first splitting lines and removing line numbers, if present. For macro actions, which consist of a predicate and one or two object arguments (e.g., *toggle-on(microwave)* or *put-on(pan, stove)*, as described in Section 2.3), the parser extracts and separates these elements. Each component is then mapped to a set of predicates and available objects taking the closest one via cosine similarity. However, this initial step does not guarantee the action's feasibility. To ensure the action is valid, we symbolically generate all possible predicate-object pairs/triples in the set of valid actions $\mathcal{A}$ and match them with the extracted pairs/triples based on semantic similarity of their sentence embeddings.

**Symbolic Validator.** To define the pre- and post-conditions in our system we use conjunctive (*and*) as well as disjunctive (*or*) and conditional (*when*) PDDL statements. We implemented an ad-hoc validator in python that is based on PDDL adapted to the specific AI2Thor environment. As mentioned in Section 2.4, we manually defined the pre- and post-conditions of the atomic actions, as they are tied to the AI2Thor simulator. To generate pre- and post-conditions for the macro actions we employ a one-shot prompt template in which the LLM is given the following elements: a short description of what formal pre- and post-conditions are, and what kind of predicates can be used to express them (e.g., predicates describing objects state, properties or physical locations, etc.); the goal-oriented task description; the plan of macro actions generated to accomplish the task; and a list of relevant objects (selected via KG-RAG, if enabled).

## 4. Experiments

The main results can be summarized as follows: (1) HVR significantly outperforms all baselines across both small and large LLMs and tasks of moderate-to-high complexity. (2) RAG is crucial for smaller LLMs, while hierarchical planning is more impactful for larger LLMs, as a longer context makes retrieval less useful. (3) Formally correct plans strongly correlate with correctly generated plans, indicat-

*Table 2.* Summary of results using Phi3 and Gemini across the metrics Plan Correctness (PC), Length Discrepancy (LD), Expanded Plan Verification (EPV), Macro Plan Verification (MPV), and Atomic Action Block Verification (AABV), averaged over all 12 tasks. PC is also averaged separately for moderate and high-complexity tasks. MPV and AABV are reported after correction (with pre-correction values in parentheses). Green indicates metrics for plan generation and execution, purple and gray (for negative values) represent plan minimality metrics, and blue indicates metrics for symbolic verification and plan correction. All results are reported as percentages.

| | | Plan Correctness | | | Length Discrepancy | | | | Verification | | |
|---|---|---|---|---|---|---|---|---|---|---|---|
| | | all | moderate | high | min | max | avg. | abs. avg. | EPV | MPV | AABV |
| **Phi3** | **HVR (our)** | 59.66 | 51.59 | 67.73 | 39.39 | 562.50 | 203.39 | 203.39 | 47.39 | (47.03) 74.20 | (27.41) 37.14 |
| | HV | 18.91 | 16.34 | 21.48 | 34.15 | 626.09 | 202.01 | 202.01 | 39.69 | (63.51) 87.86 | (6.90) 25.24 |
| | HR | 55.50 | 51.48 | 59.52 | 46.34 | 223.53 | 123.72 | 123.72 | 20.08 | - | (17.91) - |
| | VR | 20.04 | 31.08 | 9.01 | -24.39 | 262.50 | 41.59 | 58.75 | 28.57 | - | - |
| | R | 17.92 | 28.18 | 7.67 | -48.48 | 387.50 | 41.11 | 62.31 | 11.58 | - | - |
| | LLM | 11.89 | 18.92 | 4.85 | -30.43 | 116.67 | 22.61 | 36.68 | 21.11 | - | - |
| **Gemini** | **HVR (our)** | 94.19 | 100.00 | 88.39 | 33.33 | 337.50 | 109.13 | 109.13 | 88.11 | (100.00) 100.00 | (16.33) 79.83 |
| | HV | 85.27 | 93.06 | 77.48 | 21.74 | 158.82 | 67.25 | 67.25 | 100.00 | (92.11) 100.00 | (42.11) 100.00 |
| | HR | 49.01 | 68.43 | 29.60 | -50.00 | 75.00 | 32.31 | 41.40 | 25.82 | - | (29.91) - |
| | VR | 28.50 | 40.10 | 16.89 | -17.39 | 126.67 | 20.02 | 27.33 | 32.14 | - | - |
| | R | 23.87 | 37.56 | 10.18 | -27.78 | 37.50 | -4.43 | 23.20 | 15.36 | - | - |
| | LLM | 17.91 | 31.68 | 4.15 | -41.18 | 37.50 | -3.49 | 19.54 | 16.37 | - | - |

ing the effectiveness of symbolic verification in improving plan quality. (4) LLM-based planners produce unnecessarily long plans, even with RAG, hierarchical planning, and verification; corrections improve accuracy at the cost of introducing additional steps. (5) Simulator execution of correct plans does not always reach the end goal, with an average success rate of 95%, highlighting limitations in current state-of-the-art simulators. (6) LLM-based planners excel in tasks with well-defined goal states but struggle with open-ended objectives and increased task complexity.

### 4.1. Metrics and Baselines

We proposed six new metrics to evaluate the system's performance and the effectiveness of the integrated LLM.

**Plan generation and execution.** To assess the quality of both plan generation and its execution, we introduced two key metrics: (1) Plan Correctness (**PC**) identifies the furthest point in the expanded plan that aligns successfully with the ground truth (GT) plan. It is calculated as the ratio of correctly planned steps up to the first incorrect step, divided by the total number of steps in the GT-plan. (2) Execution Success (**ES**) measures how far a correctly generated plan can be executed in the environment without a failure occurring. It is defined as the ratio of steps executed correctly, up to the first failure, to the total number of steps in the GT-plan. This highlights the limitations of the simulator, as it sometimes fails to execute even entirely accurate plans.

**Plan Minimality.** To assess the length of the generated plan in comparison to the minimal ground truth plan we defined (3) Length Discrepancy (**LD**) metric which measures the discrepancy in the number of steps between the GT-plan and the generated plan. Results are reported as the average of absolute differences, signed differences, and the range of

discrepancies (minimum and maximum values). Negative values indicate that the generated plan is shorter than the GT-plan, while positive values indicate it is longer.

**Symbolic Verification and Plan Correction.** To assess the formal correctness and feasibility of the generated plans and measure the impact of symbolic corrections in improving the accuracy, we defined the following three metrics: (4) Expanded Plan Verification (**EPV**) indicates the extent to which the expanded plan (full sequence of atomic actions) has been successfully verified. It is calculated as the ratio of verified steps to the total number of steps in the generated plan. EPV is computed after all corrections have been applied. (5) Macro Plan Verification (**MPV**) measures the extent to which the macro plan has been verified. It is calculated as the ratio of verified macro plan steps to the total number of macro steps. MPV is computed in two stages: before and after any symbolic macro plan corrections. (6) Atomic Action Block Verification (**AABV**) evaluates the extent to which the macro plan has been verified at the level of atomic action blocks. It is determined by dividing the number of verified atomic action blocks by the total number of macro actions. AABV is computed both before and after corrections to the expanded plan.

**Baselines.** We evaluated our system against five baselines: (1) **HV**: Utilizes hierarchical planning with symbolic verification but without RAG (the complete KG is provided as context)[2]. (2) **HR**: Implements hierarchical planning and RAG but omits symbolic verification. (3) **VR**: Incorporates both symbolic verification and RAG, but without hierarchical planning. (4) **R**: Uses only RAG, without verification or hierarchical planning. (5) **LLM**: The LLM serves purely as a planner, with the entire KG provided as context[2].

---

[2]Only feasible in small environments with limited objects.

## 4.2. Results

We calculated the results for all metrics across all 12 tasks for all the LLMs under consideration (see Section 3.3). A summary of the results for Phi3 and Gemini is presented in Table 2. The complete results are provided in Appendix D and C, omitted here due to space constraints.

**Plan generation and execution.** The results for plan correctness in Table 2 (details in Appendix Tables 12 and 6) demonstrate that HVR significantly outperforms the other methods. For larger LLMs, HV performs better than HR, highlighting the importance of symbolic verification for more capable models. In this case, RAG appears to provide minimal benefit, likely because the larger context length of these models is sufficient to include all the necessary information. However, this may change in scenarios with larger environments containing many actions and objects, where context limitations could make RAG more valuable. Conversely, for smaller LLMs, HR outperforms HV, indicating that the RAG component is essential in this case. The combination of hierarchical decomposition, RAG, and symbolic verification, consistently outperforms all other configurations, showing that these three components complement each other effectively. Furthermore, the difference in performance gains suggest that larger LLMs are better equipped to leverage the advantages of hierarchical decomposition, symbolic verification, and RAG. However, smaller LLMs, when paired with HVR, still achieve reasonable performance. Another key observation is the difference in Plan Correctness between tasks of moderate and high complexity. Using Phi-3, the performance gap between these two categories is relatively small (32.93% vs 27.74% averaged on all the systems), whereas for Gemini, the gap is much larger (61.80% vs 38.22% averaged on all the systems) suggesting that larger LLMs are very efficient on easier tasks and struggle only on long-horizon complex tasks.

For the tasks that were planned correctly, we evaluated the execution success ES metric and found that, on average, the simulator was able to execute approximately 95% (details in Appendix Tables 13 and 7) of the planned actions. This highlights certain limitations of current simulators, such as AI2Thor, which lack the robustness needed to reliably handle complex or long-horizon tasks. Notably, this percentage is slightly lower for tasks with high complexity compared to those with moderate complexity. A likely contributing factor to these failures is the noise in the plans generated by LLM-based planners, specifically the presence of superfluous actions, which strain the simulator and increase error rates. These execution failures underscore the critical need for robust failure detection and correction methods. Our approach, which also serves as an effective failure detector (similar to other methods like (Cornelio & Diab, 2024)), offers a mechanism to identify and address such errors. These methods are not only crucial for overcoming the inherent limitations of simulators but are also indispensable for managing execution errors in real-world robotics applications.

**Plan Minimality:** We analyzed the results for the plan minimality metric (see Table 2, with details in Appendix Tables 14 and 8) and observed that smaller LLMs tend to generate significantly longer plans, with HVR producing plans that are 200% longer than the minimal ground truth plan. Larger LLMs, on the other hand, generate more efficient plans, with HVR resulting in only a 100% increase in length over the minimal plan. While the use of more powerful LLMs improves plan efficiency, our analysis shows that LLM-based planners, on average, still produce plans with redundant steps (see Appendix B.1 for examples). Interestingly, HVR increases plan length compared to the other methods, but also improves accuracy. This suggests that the corrections introduced during planning may come with the cost of adding unnecessary steps. Furthermore, in methods that do not employ hierarchical decomposition, we observed negative plan length values, indicating that the generated plans are shorter than the minimal ground truth plan. This issue is absent in methods that use hierarchical decomposition, which suggests that hierarchical decomposition helps avoiding oversimplification or omission of essential steps.

**Symbolic Verification and Plan Correction.** The results for the EPV metric (see Table 2, with details in Appendix Tables 15 and 9) demonstrate that incorporating the verifier improves both the performance and quality of the generated plans. When analyzing EPV alongside plan correctness, a strong correlation emerges, indicating that a more formally correct plan is also closer to the ground truth plan. Also examining the results with MPV and AABV metrics (see Table 2, with details in Appendix Tables 16 and 10 for MPV and Appendix Tables 17 and 11 for AABV), it is evident that plan correction leads to significant improvements, highlighting its effectiveness. The improvement in MPV is most pronounced with Phi-3, as Gemini already produces macro plans that are largely formally correct, reflecting its ability to accurately generate pre- and post-conditions. In contrast, Phi-3 struggles more with this aspect. For atomic actions blocks, achieving formal correctness is generally more challenging. However, the results show that the correction substantially enhances plan quality for both LLM models and across different methods (HV and HVR).

**Additional results**. Further results, including the performance comparison between T5 and T5bis, are provided in Appendix B.5, showing that LLM-based planners excel in tasks with well-defined goal states but struggle with open-ended objectives. Additional details, including the comparison with state-of-the-art systems, efficiency considerations, a discussion on HVR limitations and the full set of experimental results can be found in the Appendix.

## Impact Statement

This paper aims to advance Machine Learning and Robotics by improving planning in complex, dynamic environments. Our HVR method integrates LLMs with hierarchical planning, symbolic verification, and RAG to enable robots to handle long-horizon tasks by dynamically adapting plans based on real-time conditions. Such approaches are particularly crucial in specialized domains like healthcare (e.g., surgical robots), automated navigation and transportation (e.g., spacecraft or autonomous vehicles), and domestic assistance (e.g., kitchen robots), where general commonsense knowledge is insufficient.

By combining LLM flexibility with symbolic verification, HVR improves planning accuracy, reduces errors, and creates reusable macro-action libraries that allows knowledge transfer across agents. This enhances scalability, collaboration, and reliability in autonomous systems, enabling them to perform tasks with minimal human intervention by recognizing and correcting failures to ensure a safer environment for humans.

HVR is designed to tackle complex scenarios highly relevant to real-world applications. Our experiments, using freely available LLMs to ensure reproducibility, show that even newer, proprietary models could benefit from our approach, particularly for tasks requiring specialized knowledge. These work hold significant societal potential for safety-critical industries, where trust, explainability, and accountability—supported by our symbolic verification—are essential.

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

# A. Extended State of the Art Review

**LLM as planners**  Recent studies have explored the use of pre-trained Large Language Models (LLMs) for planning and executing actions in interactive environments by leveraging the prior knowledge embedded in LLMs for plan generation (Wu et al., 2023; Wei et al., 2022; Ahn et al., 2022; Yao et al., 2020; Huang et al., 2022; Schick et al., 2023). Typically, this process involves translating multi-modal observations into natural language, using the LLM to generate domain-specific actions or plans, and subsequently employing an agent to execute them. However, these methods are prone to hallucination (Golovneva et al., 2022; Pan et al., 2024) and often lack the ability for deep understanding and reasoning (Ahn et al., 2022). The rapid advancements and growing adoption of LLMs in recent years have facilitated their extensive application across diverse AI research domains, including robotics (Ren et al., 2023; Das & Chernova, 2021; Jin et al., 2024; Vemprala et al., 2023). LLMs can be directly employed for generating plans (Lin et al., 2023a; Xie, 2020; Huang et al., 2022; Carta et al., 2023; Song et al., 2023; Lin et al., 2023b). In addition to using off-the-shelf pre-trained LLMs, several studies have explored fine-tuning LLMs specifically for planning tasks (Valmeekam et al., 2023; Li et al., 2022; Pallagani et al., 2022) However, these studies indicate that LLMs are more effective at goal translation than at actual planning, as they often lack robust reasoning capabilities and are highly sensitive to prompt variations. Thus, LLMs have been utilized for translating natural language (NL) goals into symbolic languages. For instance, they have been employed to convert NL task descriptions into PDDL (Planning Domain Definition Language) representations (Guan et al., 2023; Liu et al., 2023a) allowing the problem to be solved by a standard PDDL solver. Similarly, other studies (Liang et al., 2023; Singh et al., 2023) have translated NL instructions into Python-style code, making it easier to generate executable plans for robotic tasks. Finally, (Silver et al., 2022) studied how well pretrained large language models, with few-shot prompting, perform in the role of solvers for PDDL planning problems.

**Hierarchical planning**  Hierarchical planning has recently garnered significant interest from the research community, as evidenced by the following studies: (Höller et al., 2020) introduce HDDL, an extension of PDDL (Planning Domain Definition Language) that serves as a standardized input language, enabling support for hierarchical planning tasks across various systems. HDDL aims to facilitate the comparison and integration of different hierarchical planning systems by providing a common feature set. (Ajay et al., 2024) introduce a model that integrates expert foundation models trained on language, vision, and action data to address long-horizon tasks. HiP employs an LLM for symbolic planning, a video diffusion model for visual grounding, and an inverse dynamics model to translate image trajectories into actionable steps, emphasizing iterative refinement across task planning, visual planning, and action planning levels. Unlike our symbolic approach, the hierarchy in HiP operates at the robot movement level rather than at the level of macro actions. (Wu et al., 2024) present MLDT, a method that tackles long-horizon task planning challenges by decomposing tasks into three levels: goal-level, task-level, and action-level. The approach uses three distinct prompting templates to generate plans at each level, with each stage conditioned on the output of the preceding layer. Finally, other works (Hughes et al., 2022; Rana et al., 2023) leverage the hierarchical structures of the environment by utilizing 3D scene graphs for more effective task planning in robotic systems.

**RAG systems for Robotics.**  Retrieval-Augmented Generation (RAG) (Lewis et al., 2020) systems enhance language model predictions by retrieving task-specific relevant knowledge from external sources such as text (Guu et al., 2020), knowledge graphs (Edge et al., 2024), or web (Kim et al., 2024). RAG has become increasingly important for generative models, especially given recent findings that indicate LLMs are becoming less reliable (Zhou et al., 2024a). (Goyal et al., 2022) explore retrieval in the context of deep reinforcement learning (RL) agents, though it does not employ LLMs, which limits its applicability and efficiency. Other studies emphasize the retrieval of past experiences to support planning processes. For instance, (Xu et al., 2024) focus on retrieving plans related to similar tasks, demonstrating the value of retrieval in supporting planning processes. Similarly, (Jain et al., 2024) enhances LLMs as policies by retrieving relevant past interactions from a continuously expanding database, which stores both previous interactions and feedback from critics; (Zhu et al., 2024a) utilize a policy retriever to extract robotic policies from a large-scale policy memory; and (Kagaya et al., 2024) employ retrieval of past experiences from a memory structure to inform agent planning. Examples of Graph-RAG methods for planning include the work of (Rana et al., 2023) which performs a semantic search for task-relevant subgraphs from a condensed representation of a full scene graph representing the environment, optimizing the planning process; and the work of (Zhu et al., 2024b) which introduces KNOWAGENT, an method that enhances LLM planning capabilities by incorporating explicit knowledge retrieved from an action knowledge base and applying a knowledgeable self-learning strategy to guide action paths during planning. Finally, the most relevant work to ours is the work of (Lu et al., 2023) which utilizes a KG-RAG method for retrieving relevant nodes from a knowledge graph (KG) while mapping the actions to only

feasible ones. However, their approach lacks a comprehensive verification of the entire plan and is likely to encounter the same challenges as LLMs when handling complex, long-horizon tasks.

**Symbolic verification and LLMs**   The position paper by  (Kambhampati et al.) argues that LLMs are not suitable as standalone planners or plan verifiers, and instead proposes a framework that combines LLMs with external verifiers, named *LLM-Modulo Framework*, with a particular focus on PDDL planning tasks. Several implementations of this concept exist. For instance, FunSearch (Romera-Paredes et al., 2024) and Alpha-Geometry (Trinh et al., 2024) employs a alternate between a fine-tuned LLM that generates potential solutions and an a symbolic evaluator that evaluate them.  In reinforcement learning scenarios with simulators, the simulator itself serves as plan evaluator and critique mechanism within the LLM-Modulo framework.  For instance, in the works of (Rajvanshi et al., 2024) and (Ahn et al., 2022) the simulator help to explicitly filter the action choices suggested by the LLM. Another relevant work is CLAIRIFY (Skreta et al., 2023), which combines iterative prompting with program verification to ensure that the generated programs are syntactically correct and incorporate environmental constraints. However, CLAIRIFY's verifier only checks the syntax of instructions and the existence of hardware and reagents used (thereby avoiding hallucinations), without validating the overall correctness of the plan. Similarly, another work (Raman et al., 2022) identifies and corrects errors in LLM-generated plans through iterative re-prompting using the error information. However, this approach is limited to handling preconditions, and it assumes that error information is provided (e.g., generated by a simulator) rather than detected independently. Most relevant for our approach is the work by (Valmeekam et al., 2023), which demonstrates that LLM performance improves significantly when backprompting with the VAL-verifier (Howey et al., 2004). This highlights the effectiveness of integrating verification tools in conjunction with LLM-based planning.

## B. Additional details

### B.1. Examples of alternative plans vs the ground truth plans

As discussed in the paper, generated plans often tend to be longer and include unnecessary steps, which can affect execution success even when the plan is correct leading to the desired goal.  This is partly due to LLM-based planners generating redundant actions during task decomposition and error correction. For example, in task T3, where the goal is "I want a fried egg in a plate on the countertop", the plan generated by HVR includes several superfluous steps that do not contribute to achieving the objective efficiently.

```
 1  ALTERNATIVE PLAN (32 steps)              GROUD THRUTH PLAN (16 steps)
 2  (navigate_to_obj, Egg-1)                 (navigate_to_obj, Egg-1)
 3  (pick_up, Egg-1)                         (pick_up, Egg-1)
 4  (navigate_to_obj, Egg-1)                 (crack_obj, Egg-1)
 5  (crack_obj, Egg-1)                       (navigate_to_obj, Pan-1)
 6  (navigate_to_obj, Pan-1)                 (put_in, EggCracked, Pan-1)
 7  (navigate_to_obj, CounterTop-1)          (pick_up, Pan-1)
 8  (put_on, EggCracked-1, CounterTop-1)     (navigate_to_obj, StoveBurner-1)
 9  (navigate_to_obj, Pan-1)                 (put_on, Pan-1, StoveBurner-1)
10  (pick_up, Pan-1)                         (toggle_on, StoveBurner-1)
11  (navigate_to_obj, StoveBurner-1)         (toggle_off, StoveBurner-1)
12  (put_on, Pan-1, StoveBurner-1)           (pick_up, EggCracked-1)
13  (navigate_to_obj, StoveBurner-1)         (navigate_to_obj, Plate-1)
14  (toggle_on, StoveBurner-1)               (put_on, EggCracked-1, Plate-1)
15  (navigate_to_obj, EggCracked-1)          (pick_up, Plate-1)
16  (pick_up, EggCracked-1)                  (navigate_to_obj, CounterTop-1)
17  (navigate_to_obj, Pan-1)                 (put_on, Plate-1, CounterTop-1)
18  (put_in, EggCracked-1, Pan-1)
19  (navigate_to_obj, StoveBurner-1)
20  (toggle_off, StoveBurner-1)
21  (navigate_to_obj, Plate-1)
22  (pick_up, Plate-1)
23  (navigate_to_obj, EggCracked-1)
24  (navigate_to_obj, CounterTop-1)
25  (put_on, Plate-1, CounterTop-1)
26  (navigate_to_obj, EggCracked-1)
27  (pick_up, EggCracked-1)
28  (navigate_to_obj, Plate-1)
```

```
29  (put_on, EggCracked-1, Plate-1)
30  (navigate_to_obj, Plate-1)
31  (pick_up, Plate-1)
32  (navigate_to_obj, CounterTop-1)
33  (put_on, Plate-1, CounterTop-1)
```

## B.2. Generic tasks

Tasks T12 and T5bis are open-ended, causing their ground truth plans to vary across runs based on the LLM's intentions. The goal for T12 is "Set the table and serve a vegan meal", while for T5bis, it is "I want warm water in a cup." In contrast, all other tasks have a single manually created ground truth plan, as their goals can be achieved in a unique way with a minimal number of actions. Since T12 and T5bis allow multiple ways to accomplish their goals, we generated a ground truth plan for each run, aligning it with the LLM's specific intentions.

Two examples of two ground truth plans (aligned with their corresponding generative ones) for T5bis are:

```
1   PLAN 1 (18 steps)                      PLAN-2 (17 steps)
2   navigate_to_obj(Pot-1)                 navigate_to_obj(Cup-1)
3   pick_up(Pot-1)                         pick_up(Cup-1)
4   navigate_to_obj(SinkBasin-1)           navigate_to_obj(SinkBasin-1)
5   put_in(Pot-1,SinkBasin-1)              put_in(Cup-1,SinkBasin-1)
6   toggle_on(Faucet-1)                    toggle_on(Faucet-1)
7   toggle_off(Faucet-1)                   toggle_off(Faucet-1)
8   pick_up(Pot-1)                         navigate_to_obj(Microwave-1)
9   navigate_to_obj(StoveBurner-1)         open_obj(Microwave-1)
10  put_on(Pot-1,StoveBurner-1)            navigate_to_obj(SinkBasin-1)
11  toggle_on(StoveKnob-1)                 pick_up(Cup-1)
12  toggle_off(StoveKnob-1)                navigate_to_obj(Microwave-1)
13  pick_up(Pot-1)                         put_in(Cup-1,Microwave-1)
14  navigate_to_obj(Cup-1)                 close_obj(Microwave-1)
15  pour(Pot-1,Cup-1)                      toggle_on(Microwave-1)
16  navigate_to_obj(CounterTop-1)          toggle_off(Microwave-1)
17  put_on(Pot-1,CounterTop-1)             open_obj(Microwave-1)
18  navigate_to_obj(Cup-1)                 pick_up(Cup-1)
19  pick_up(Cup-1)
```

Defining the ground truth plans is essential for computing the metrics.

## B.3. Additional Implementation Details

The code will be made available upon publication.

Here are some additional details about the implementation for reproducibility:

- The plan correctness (PC) metric is adjusted for plans that follow a partial order, where actions can be executed in parallel or in different sequences. For example, when preparing a salad with lettuce and tomato, the tomato can be cut and added first, or the lettuce can be prepared first—both sequences achieve the same goal and are valid plan linearizations. To account for this, the metric is adapted to take the maximum score across all possible linearizations.

- In the prompt for macro action generation, we provide the classes of the available object (e.g., *Apple*) to inform the LLM of what is present in the environment. In contrast, the prompt for AA-blocks expansion we list specific object instances (e.g., *apple-1*).

- When correcting an AA-block, for each step, the system attempts correction up to $2 * x$ times, where $x$, the lenght of the action block dynamically updated based on the current number of steps. For example, starting with 5 steps ($x = 5$), if a correction added a missing step, $x$ increases to 6. Similarly, $x$ adjusts whenever steps are removed. To prevent an infinite loop of corrections, each block has a static upper limit of $50$ steps.

- For macro actions, an initial attempt is made to correct pre- and post-conditions, based on the symbolic validator feedback, followed by further corrections after each AA-block execution. At the end of an AA-block execution, if

no failures occur, the system reviews and refines the pre- and post-conditions of the macro action that generated it, incorporating feedback from the environment. This refinement allows for updating and saving a better quality of macro action in the macro-action "culture" library.

### B.4. Task renumbering

In Table 3 we report the task renumbering we used in comparison with RECOVER (Cornelio & Diab, 2024). Task "Boil water in a pot" from (Cornelio & Diab, 2024) was used as a one-shot example in our LLM-based planner. Additionally, we introduced two new complex tasks, T11 and T12. Task T5 was duplicated as T5bis, with the difference that in T5, the robot must warm water specifically using the microwave, whereas in T5bis, the robot can warm the water using any available method, such as on a stove, in a kettle, or in the microwave.

*Table 3.* Task numbering: our work vs RECOVER (Cornelio & Diab, 2024) (*used as 1-shot example)

| | HVR ID | RECOVER ID | Name |
|---|---|---|---|
| Moderate complexity | T1 | T1 | Serve wine |
| | T2 | T2 | Make coffee |
| | –* | T3 | Boil water in a pot |
| | T3 | T4 | Fry egg in a pan |
| | T4 | T5 | Toast bread |
| | T5 | T6 | Warm water (in microwave) |
| | T5bis | - | Warm water (generic) |
| | T6 | T7 | Cook potato slice (in microwave) |
| High complexity | T7 | T8 | Simple salad |
| | T8 | T10 | Vegan sandwich |
| | T9 | T11 | Cook egg and potato slice |
| | T10 | T12 | Complex salad |
| | T11 | – | Tomato-egg toast |
| | T12 | – | Complex plate |

### B.5. Additional results: T5 vs T5bis

Table 4 presents the results comparing T5 "Warm water in a cup using the microwave" (specific goal) and T5bis "Warm water in a cup" (general goal) across various metrics. The results show that tasks with more general objectives lead to worse performance due to increased ambiguity in goal interpretation due to fact that there are multiple ways to accomplish it. This highlights the challenge LLM-based planners face in handling open-ended tasks.

*Table 4.* Results comparing T5 and T5bis over the different metrics Plan Correctness (PC), Length Discrepancy (LD), Expanded Plan Verification (EPV), Macro Plan Verification (MPV) after correction, and Atomic Action Block Verification (AABV) after correction.

| | Gemini | | | | | | Phi3 | | | | | |
|---|---|---|---|---|---|---|---|---|---|---|---|---|
| | PC | | LD | | EPV | | PC | | LD | | EPV | |
| | T5bis | T5 | T5bis | T5 | T5bis | T5 | T5bis | T5 | T5bis | T5 | T5bis | T5 |
| **HVR (our)** | 88.89 | 100.00 | 100.00 | 223.53 | 96.88 | 34.55 | 23.53 | 29.41 | 368.75 | 335.29 | 22.67 | 6.76 |
| HV | 52.94 | 88.24 | 175.00 | 158.82 | 100.00 | 100.00 | 0.00 | 11.76 | 868.75 | 305.88 | 5.16 | 11.59 |
| HR | 29.41 | 29.41 | 37.50 | 5.88 | 12.20 | 13.16 | 27.78 | 17.65 | 268.75 | 223.53 | 2.56 | 6.67 |
| VR | 17.65 | 52.94 | 37.50 | 23.53 | 7.32 | 51.22 | 11.76 | 17.65 | 125.00 | 11.76 | 5.45 | 46.15 |
| R | 17.65 | 52.94 | 25.00 | -17.65 | 7.69 | 20.59 | 11.76 | 11.76 | 0.00 | -5.88 | 8.57 | 8.33 |
| LLM | 17.65 | 17.65 | 0.00 | -41.18 | 8.57 | 10.00 | 0.00 | 11.76 | 162.50 | 5.88 | 32.79 | 23.68 |
| Avg. | 0.32 | 0.49 | 62.50 | 58.82 | 38.78 | 38.25 | 0.11 | 0.14 | 298.96 | 146.08 | 12.87 | 17.20 |

### B.6. Library of Macro Actions

ONTOTHOR (Cornelio & Diab, 2024) contains the class **Action** to represent the agent's interactions with the environment, such as object manipulations and environmental observation. We further refined this structure by introducing two subclasses:

**Atomic Action**, which aligns with the original **Action** class in ONTOTHOR, and a newly defined **Macro Action** class, used to represent higher-level operations like *boil-water*.

During task execution, instances of both Atomic Actions (AAs) and Macro Actions (MAs) are stored in the agent's Knowledge Graph $\mathcal{G}$. Each **Macro Action** instance is connected to its natural language description—generated by the LLM and including pre- and post-conditions—via the *hasDescription* predicate. These pre- and post-conditions could also be modeled as triples, extending the scene-graph representation introduced by Cornelio & Diab (2024). Additionally, each Macro Action instance is linked to its corresponding sequence of Atomic Actions using the *hasAtomicAction* predicate.

Thus, after task execution, the Knowledge Graph $\mathcal{G}$ contains a set of macro actions along with their associated pre- and post-conditions. If a macro action is successfully verified by the environment state, it is then transferred to the ontology $\mathcal{O}$. For details on how this alignment between observations and the expected world state is verified after each macro action, refer to Section 2.4.

### B.7. Efficiency Considerations

Figure 4 shows the average computational time for the different models across the 13 tasks using Gemini-1.5-flash as LLM. As expected, approaches involving hierarchical decomposition require (approximately three times) more time than those without. However, as LLMs become faster, the overhead introduced by the HVR framework becomes increasingly negligible, while substantially improving plan correctness. For example, running the same tasks with Gemini-2.0-flash reduced the average execution time for HVR from 3285.32 seconds to 681.51 seconds—a 5x improvement.

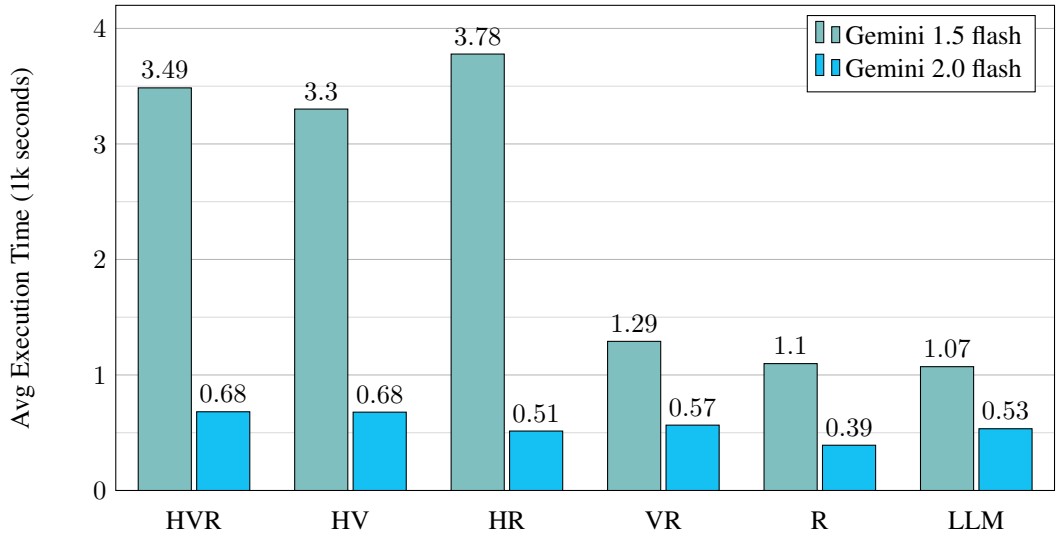

*Figure 4.* Average execution times (in seconds) per model using Gemini 1.5 and 2.0 flash

In contrast to this improvement, relying solely on an LLM as a planner demands a substantially larger context window, which quickly becomes impractical as task or environment complexity increases. HVR addresses this limitation through retrieval-augmented generation (RAG), enabling it to dynamically access only the relevant information from the knowledge graph. This design ensures stable processing times and scalability, even as the environment grows, whereas LLM-only approaches face escalating computational demands and eventual context window exhaustion.

Figure 5 shows the trade-off between plan correctness and execution time across the different methods. While HVR takes longer to compute plans, it consistently achieves the highest correctness. In contrast, simpler methods like LLM or R achieve faster execution but produce significantly less reliable plans, demonstrating the benefit of HVR's more structured approach.

### B.8. Comparison with the State-of-the-Art

We do not provide a direct experimental comparison with existing state-of-the-art methods, as we were unable to find prior work that targets complex, long-horizon tasks in a kitchen environment compatible with our setup.

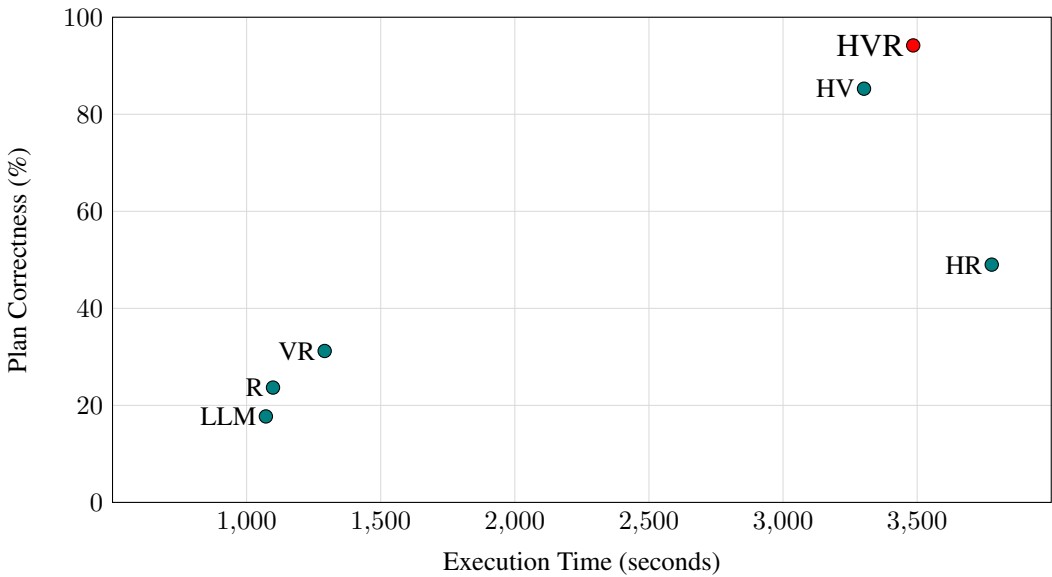

*Figure 5.* Efficiency of Methods: Plan Correctness vs. Execution Time in seconds with Gemini 1.5 flash

However, we conducted comparison experiments[3] with two relevant approaches: Smart-LLM and ProgPrompt (see details below). While we also considered several other methods, we were unable to include them in our evaluation due to compatibility issues (see below for more detailed information).

**Smart-LLM** (Kannan et al., 2024) uses large language models for planning in robotic systems, with a focus on multi-agent coordination. It handles task decomposition, coalition formation, and task allocation among heterogeneous robot teams based on their capabilities. The approach focuses on generating executable Python code tailored to the robot skills and environment, with an emphasis on parallel and distributed execution across agents.

This work is closely related to ours, as it uses the same simulator (AI2Thor) for the experiments and addresses complex tasks (even if in a multi-agent setting).

In the `data/final_test` folder in their GIT repository there are 7 rooms of which only 3 are kitchen: `FloorPlan6`, `FloorPlan15`, `FloorPlan21`. The others are bedrooms and living rooms which are outside the scope of our ontology. There are a total of 15 tasks in these 3 rooms of which only 12 were feasible in our setup: 3 tasks out of 3 for `FloorPlan6`; 5 tasks out of 6 for `FloorPlan15`; and 4 tasks out of 6 for `FloorPlan21`. This is due to a key implementation choice: in Smart-LLM, the robot has full knowledge of all object locations, including hidden ones (e.g., inside a fridge or cupboard), whereas in our system, consistently with prior work like RECOVER (Cornelio & Diab, 2024), we assume only visible objects can be used — interacting with hidden objects is thus treated as failure.

We evaluated HVR on these 12 tasks, and it successfully planned and executed all of them.

**ProgPrompt** (Singh et al., 2023) uses large language models for robotic task planning, with a focus on generating structured, Python-like programs that include pre/post condition in the form of assertions and recovery steps. This helps ensure the plans are grounded and executable. The method relies on careful prompt engineering, incorporating imports, object lists, and example functions to guide the model's output.

In the `progprompt-vh/data/test_unseen/` folder of their GIT repository, there are 10 available tasks (see Table 2in Singh et al. (2023)), 7 of which are kitchen related. We excluded the following tasks as they are either human-specific or fall outside the scope of our ontology: *brush teeth*; *watch tv*; and *eat chips on the sofa*. Although these tasks were originally designed for the VirtualHome simulator, we reimplemented them in the AI2Thor simulator. Table 5 provides details of this reimplementation.

---

[3]For these additional experiments we used Gemini 2.0 flash, since these experiemnts were done in a subsequent moment.

*Table 5.* ProgPrompt tasks translated from VirtualHome simulator to AI2Thor simulator.

| VirtualHome Description | AI2Thor description | AI2Thor room |
| --- | --- | --- |
| put salmon in the fridge | put *egg* in the fridge | FloorPlan2 |
| wash the plate | wash the plate | FloorPlan16 |
| bring coffeepot and cupcake to the coffee table | bring *kettle* and *apple* to the *table* | FloorPlan18 |
| microwave salmon | microwave *potato* | FloorPlan2 |
| turn off light | turn off light | FloorPlan1 |
| throw away apple | throw away apple | FloorPlan1 |
| make toast | make toast | FloorPlan1 |

HVR was able to plan and execute correctly all of the tasks. However, it's worth noting that all of these tasks are relatively simple and fall well below the complexity of those in our benchmark. Additionally, ProgPrompt was evaluated using GPT-3, and their performance would likely improve with a more capable LLM.

**Other works we considered.** We also considered other LLM-based systems such as Liu et al. (2023a) and Guan et al. (2023). However, these methods are built on different simulators and/or involve tasks that are not compatible with our setup (e.g., organizing blocks on a table). As a result, a direct comparison would require a substantial and non-trivial reimplementation, which falls outside the scope of this work.

Similarly, we could not compare with PDDL-based planning systems *on our tasks* as we found no existing works using planners expressive enough to support our tasks. As described in Section 3.3 ('Symbolic Validator'), our system uses conjunctive (*and*), disjunctive (*or*), and conditional (*when*) PDDL statements. We therefore implemented a custom validator in Python, adapted specifically to the AI2Thor kitchen environment.

We have considered the implementation of Zhou et al. (2024b), which involves solving simple abstract tasks in a simulated kitchen environment (e.g., organizing generic 'ingredient' objects in 'pot' objects by following a 'recipe'). In this work, the authors generate both a PDDL domain and a goal state starting from a task specification. While this can work for a simple environment with few available actions, we argue it is generally not possible to generate PDDL domain specifications from high-level natural language task specifications such as the ones we use in our work. Thus, we did not run experimental comparisons with this method.

Finally, we reviewed the implementation of Xie (2020) which uses the ALFRED simulator, a close relative of AI2Thor. However, the implementation is too limited for our use case and does not support key functionalities like creating object slices after slicing an object, handling liquids, or modeling interactions such as opening/closing appliances, which are necessary to create plans for the vast majority of our tasks. The actions modeled in the implementation are more coarse-grained than the ones we used in our work: for example, instead of modeling the act of opening a microwave, the system simply assumes that an object becomes hot when the robot is near the appliance, without simulating the intermediate steps. For these reasons, the system is not suitable for a meaningful comparison.

### B.9. Limitations and Future Work

The HVR framework integrates hierarchical planning, knowledge graph retrieval, and symbolic validation to enhance LLM-based task planning, but it also presents several limitations that point to promising directions for future work. One key limitation is its reliance on a fixed ontology and action space, which constrains generalization to new environments or tasks. While updating to the ontology currently requires expert domain knowledge, the predefined action space could be extended to include novel actions automatically using LLM-based techniques—similar to how HVR currently generates macro actions along with their corresponding pre- and post-conditions. Prompt sensitivity is another concern common to LLM-based systems, although recent models show improved robustness to variations in prompt phrasing. Additionally, HVR's current use of natural language to mediate interactions between the LLM and the knowledge graph may not be the most efficient; leveraging sub-symbolic or embedded representations could improve this integration. Another limitation is the current restriction to a linear structure for the generated plans. Supporting partial-order plans would not only allow for more flexible planning but also make it possible to execute different branches in parallel, which is especially valuable in multi-agent settings. Finally, in our method, error correction is done independently for each part of the plan and does not address interdependencies between errors at different planning levels. A more connected correction strategy that reasons

across macro and atomic actions could help improve overall correctness.

## C. Full Results using Phi3

In Table 6 we report the results for the metric Plan Correctness (PC) for HVR and all the baselines approaches using Phi3 as LLM-based planner. The results for each method are presented per task, along with the averages across all 12 tasks, as well as separately for tasks of moderate and high complexity.

*Table 6.* Results for the Plan Correctness (PC) metric using Phi3 across all 12 tasks are presented as percentages. The results are provided as overall averages and also separately averaged for moderate and high-complexity tasks.

| | Moderate complexity tasks | | | | | | High complexity tasks | | | | | | | | |
| | T1 | T2 | T3 | T4 | T5 | T6 | T7 | T8 | T9 | T10 | T11 | T12 | avg. | avg. moderate | avg. high |
|---|---|---|---|---|---|---|---|---|---|---|---|---|---|---|---|
| **HVR (our)** | 37.50 | 100.00 | 31.25 | 72.22 | 29.41 | 39.13 | 100.00 | 100.00 | 25.71 | 69.44 | 51.22 | 60.00 | 59.66 | 51.59 | 67.73 |
| HV | 37.50 | 0.00 | 0.00 | 44.44 | 11.76 | 4.35 | 20.69 | 18.18 | 8.57 | 16.67 | 53.66 | 11.11 | 18.91 | 16.34 | 21.48 |
| HR | 37.50 | 100.00 | 31.25 | 83.33 | 17.65 | 39.13 | 82.76 | 84.85 | 17.14 | 69.44 | 48.78 | 54.17 | 55.50 | 51.48 | 59.52 |
| VR | 25.00 | 33.33 | 31.25 | 44.44 | 17.65 | 34.24 | 10.34 | 24.24 | 5.71 | 5.56 | 4.88 | 3.33 | 20.04 | 31.08 | 9.01 |
| R | 25.00 | 100.00 | 12.50 | 11.11 | 11.76 | 8.70 | 13.79 | 6.06 | 5.71 | 5.56 | 4.88 | 10.00 | 17.92 | 28.18 | 7.67 |
| LLM | 25.00 | 44.44 | 12.50 | 11.11 | 11.76 | 8.70 | 6.90 | 6.06 | 5.71 | 5.56 | 4.88 | 0.00 | 11.89 | 18.92 | 4.85 |
| | | | | | | | | | | | | | 30.65 | 32.93 | 28.38 |

In Table 7 we report the results for the metric Execution Success (ES) for HVR and all the baselines using Phi3 as LLM-based planner. The results for each method are presented per task that was planned correctly (100% Plan Correctness).

*Table 7.* Results for the Execution Success (ES) metric using Phi3 across all 12 tasks are presented as percentages.

| | Moderate complexity tasks | | | | | | High complexity tasks | | | | | |
| | T1 | T2 | T3 | T4 | T5 | T6 | T7 | T8 | T9 | T10 | T11 | T12 |
|---|---|---|---|---|---|---|---|---|---|---|---|---|
| **HVR (our)** | - | 100.00 | - | - | - | - | 86.20 | 100.00 | - | - | - | - |
| HV | - | - | - | - | - | - | - | - | - | - | - | - |
| HR | - | 100.00 | - | - | - | - | - | - | - | - | - | - |
| VR | - | - | - | - | - | - | - | - | - | - | - | - |
| R | - | 100.00 | - | - | - | - | - | - | - | - | - | - |
| LLM | - | - | - | - | - | - | - | - | - | - | - | - |

In Table 8 we report the results for the metric Length Discrepancy (LD) for HVR and all the baselines using Phi3 as LLM-based planner. The results for each method are presented per task, along with the average across all 12 tasks, the absolute average, the min and max values.

*Table 8.* Results for the Length Discrepancy (LD) metric using Phi3 across all 12 tasks are presented as percentages. The results include averages, absolute averages, and the minimum and maximum values across tasks.

| | Moderate complexity tasks | | | | | | High complexity tasks | | | | | | | | |
| | T1 | T2 | T3 | T4 | T5 | T6 | T7 | T8 | T9 | T10 | T11 | min | max | avg | abs. avg. |
|---|---|---|---|---|---|---|---|---|---|---|---|---|---|---|---|
| **HVR (our)** | 100.00 | 222.22 | 562.50 | 266.67 | 335.29 | 256.52 | 58.62 | 39.39 | 240.00 | 100.00 | 56.10 | 39.39 | 562.50 | 203.39 | 203.39 |
| HV | 587.50 | 166.67 | 106.25 | 50.00 | 305.88 | 626.09 | 41.38 | 121.21 | 85.71 | 97.22 | 34.15 | 34.15 | 626.09 | 202.01 | 202.01 |
| HR | 100.00 | 144.44 | 200.00 | 155.56 | 223.53 | 134.78 | 79.31 | 69.70 | 160.00 | 47.22 | 46.34 | 46.34 | 223.53 | 123.72 | 123.72 |
| VR | 262.50 | 166.67 | 25.00 | 55.56 | 11.76 | 30.43 | -24.14 | -12.12 | -14.29 | -19.44 | -24.39 | -24.39 | 262.50 | 41.59 | 58.75 |
| R | 387.50 | 144.44 | 12.50 | -11.11 | -5.88 | 4.35 | -37.93 | -48.48 | 20.00 | -8.33 | -4.88 | -48.48 | 387.50 | 41.11 | 62.31 |
| LLM | 50.00 | -22.22 | 62.50 | 116.67 | 5.88 | -30.43 | 34.48 | 39.39 | 17.14 | -2.78 | -21.95 | -30.43 | 116.67 | 22.61 | 36.68 |

In Table 9 we report the results for the metric Expanded Plan Verification (EPV) for HVR and all the baselines using Phi3 as LLM-based planner. The results for each method are presented per task, along with the average across all 12 tasks.

*Table 9.* Results for the Expanded Plan Verification (EPV) metric using Phi3 across all 12 tasks are presented as percentages, with task averages also provided.

| | Moderate complexity tasks | | | | | | High complexity tasks | | | | | | |
| | T1 | T2 | T3 | T4 | T5 | T6 | T7 | T8 | T9 | T10 | T11 | T12 | avg. |
|---|---|---|---|---|---|---|---|---|---|---|---|---|---|
| **HVR (our)** | 93.75 | 6.76 | 21.95 | 9.43 | 56.06 | 100.00 | 15.22 | 100.00 | 11.76 | 53.13 | 44.44 | 56.16 | 47.39 |
| HV | 61.82 | 11.59 | 5.99 | 100.00 | 100.00 | 45.83 | 26.83 | 32.88 | 12.31 | 49.09 | 25.35 | 4.55 | 39.69 |
| HR | 55.56 | 6.67 | 16.25 | 8.96 | 20.90 | 2.94 | 8.33 | 45.65 | 8.53 | 27.88 | 6.52 | 32.80 | 20.08 |
| VR | 35.00 | 46.15 | 53.57 | 12.82 | 57.14 | 25.00 | 14.81 | 20.00 | 11.76 | 4.00 | 36.76 | 25.81 | 28.57 |
| R | 28.00 | 8.33 | 6.00 | 29.73 | 8.11 | 26.47 | 10.00 | 5.66 | 3.75 | 3.61 | 4.17 | 5.08 | 11.58 |
| LLM | 13.04 | 23.68 | 7.14 | 6.67 | 28.33 | 42.11 | 18.31 | 3.66 | 3.80 | 5.26 | 4.05 | 97.22 | 21.11 |

In Table 10 we report the results for the metric Macro Plan Verification (MPV) for all the methods that use symbolic validation (i.e., HVR and HV) using Phi3 as LLM-based planner. The results for each method are presented per task, along with the average across all 12 tasks. They are shown both before correction (first part of the table) and after correction (second part of the table).

*Table 10.* Results for the Macro Plan Verification (MPV) metric using Phi3 across all 12 tasks are presented as percentages. The results are shown both before and after correction, with task averages also provided.

| | Moderate complexity tasks | | | | | | High complexity tasks | | | | | | |
| | T1 | T2 | T3 | T4 | T5 | T6 | T7 | T8 | T9 | T10 | T11 | T12 | avg. |
|---|---|---|---|---|---|---|---|---|---|---|---|---|---|
| | | | | | | Before correction | | | | | | | |
| **HVR (our)** | 100.00 | 60.00 | 0.00 | 66.67 | 33.33 | 100.00 | 14.29 | 100.00 | 14.29 | 25.00 | 28.57 | 22.22 | 47.03 |
| HV | 100.00 | 100.00 | 66.67 | 100.00 | 100.00 | 33.33 | 40.00 | 40.00 | 25.00 | 28.57 | 28.57 | 100.00 | 63.51 |
| | | | | | | After correction | | | | | | | |
| **HVR (our)** | 100.00 | 60.00 | 50.00 | 66.67 | 44.44 | 100.00 | 85.71 | 100.00 | 71.43 | 37.50 | 85.71 | 88.89 | 74.20 |
| HV | 100.00 | 100.00 | 100.00 | 100.00 | 100.00 | 100.00 | 80.00 | 60.00 | 100.00 | 85.71 | 28.57 | 100.00 | 87.86 |

In Table 11 we report the results for the metric Atomic Action Block Verification (AABV) for all the methods that use symbolic validation (i.e., HVR and HV) using Phi3 as LLM-based planner. The results for each method are presented per task, along with the average across all 12 tasks. They are shown both before correction (first part of the table) and after correction (second part of the table). For comparison purposes, we also included results for HR in the first part of the table.

*Table 11.* Results for the Atomic Action Block Verification (AABV) metric using Phi3 across all 12 tasks are presented as percentages. The results are shown both before and after correction, with task averages also provided.

| | Moderate complexity tasks | | | | | | High complexity tasks | | | | | | |
| | T1 | T2 | T3 | T4 | T5 | T6 | T7 | T8 | T9 | T10 | T11 | T12 | avg. |
|---|---|---|---|---|---|---|---|---|---|---|---|---|---|
| | | | | | | Before correction | | | | | | | |
| **HVR (our)** | 50.00 | 0.00 | 0.00 | 66.67 | 0.00 | 14.29 | 0.00 | 100.00 | 0.00 | 25.00 | 28.57 | 44.44 | 27.41 |
| HV | 40.00 | 0.00 | 0.00 | 0.00 | 0.00 | 0.00 | 0.00 | 0.00 | 0.00 | 14.29 | 28.57 | 0.00 | 6.90 |
| HR | 50.00 | 0.00 | 0.00 | 14.29 | 0.00 | 14.29 | 0.00 | 71.43 | 0.00 | 0.00 | 28.57 | 36.36 | 17.91 |
| | | | | | | After correction | | | | | | | |
| **HVR (our)** | 50.00 | 100.00 | 16.67 | 66.67 | 0.00 | 14.29 | 0.00 | 100.00 | 0.00 | 25.00 | 28.57 | 44.44 | 37.14 |
| HV | 40.00 | 0.00 | 100.00 | 100.00 | 0.00 | 0.00 | 0.00 | 20.00 | 0.00 | 14.29 | 28.57 | 0.00 | 25.24 |

## D. Full Results using Gemini

In Table 12 we report the results for the metric Plan Correctness (PC) for HVR and all the baselines approaches using Gemini as LLM-based planner. The results for each method are presented per task, along with the averages across all 12 tasks, as well as separately for tasks of moderate and high complexity.

*Table 12.* Results for the Plan Correctness (PC) metric using Gemini across all 12 tasks are presented as percentages. The results are provided as overall averages and also separately averaged for moderate and high-complexity tasks.

| | Moderate complexity tasks | | | | | | High complex tasks | | | | | | | | |
| | T1 | T2 | T3 | T4 | T5 | T6 | T7 | T8 | T9 | T10 | T11 | T12 | avg. | avg. moderate | avg. high |
|---|---|---|---|---|---|---|---|---|---|---|---|---|---|---|---|
| **HVR (our)** | 100.00 | 100.00 | 100.00 | 100.00 | 100.00 | 100.00 | 100.00 | 100.00 | 57.14 | 100.00 | 73.17 | 100.00 | 94.19 | 100.00 | 88.39 |
| HV | 87.50 | 100.00 | 100.00 | 100.00 | 88.24 | 82.61 | 100.00 | 100.00 | 57.14 | 69.44 | 68.29 | 70.00 | 85.27 | 93.06 | 77.48 |
| HR | 50.00 | 100.00 | 100.00 | 83.33 | 29.41 | 47.83 | 100.00 | 0.00 | 5.71 | 69.44 | 2.44 | 0.00 | 49.01 | 68.43 | 29.60 |
| VR | 37.50 | 100.00 | 12.50 | 33.33 | 52.94 | 4.35 | 13.79 | 9.09 | 8.57 | 38.89 | 2.44 | 28.57 | 28.50 | 40.10 | 16.89 |
| R | 50.00 | 100.00 | 12.50 | 5.56 | 52.94 | 4.35 | 13.79 | 3.03 | 25.71 | 2.78 | 2.44 | 13.33 | 23.87 | 37.56 | 10.18 |
| LLM | 50.00 | 100.00 | 12.50 | 5.56 | 17.65 | 4.35 | 13.79 | 3.03 | 2.86 | 2.78 | 2.44 | 0.00 | 17.91 | 31.68 | 4.15 |
| | | | | | | | | | | | | | 49.79 | 61.80 | 37.78 |

In Table 13 we report the results for the metric Execution Success (ES) for HVR and all the baselines using Gemini as LLM-based planner. The results for each method are presented per task that was planned correctly (100% Plan Correctness).

*Table 13.* Results for the Execution Success (ES) metric using Gemini across all 12 tasks are presented as percentages.

| | Moderate complexity tasks | | | | | | High complex tasks | | | | | |
| | T1 | T2 | T3 | T4 | T5 | T6 | T7 | T8 | T9 | T10 | T11 | T12 |
|---|---|---|---|---|---|---|---|---|---|---|---|---|
| **HVR (our)** | 100.00 | 100.00 | 100.00 | 100.00 | 100.00 | 100.00 | 100.00 | 100.00 | - | 88.89 | - | 100.00 |
| HV | - | 100.00 | 100.00 | 100.00 | - | - | 86.20 | 100.00 | - | - | - | - |
| HR | - | 100.00 | 100.00 | - | - | - | 86.20 | - | - | - | - | - |
| VR | - | 100.00 | - | - | - | - | - | - | - | - | - | - |
| R | - | 100.00 | - | - | - | - | - | - | - | - | - | - |
| LLM | - | 100.00 | - | - | - | - | - | - | - | - | - | - |

In Table 14 we report the results for the metric Length Discrepancy (LD) for HVR and all the baselines using Gemini as LLM-based planner. The results for each method are presented per task, along with the average across all 12 tasks, the absolute average, the min and max values.

*Table 14.* Results for the Length Discrepancy (LD) metric using Gemini across all 12 tasks are presented as percentages. The results include averages, absolute averages, and the minimum and maximum values across tasks.

| | Moderate complexity tasks | | | | | | High complexity tasks | | | | | | | | |
| | T1 | T2 | T3 | T4 | T5 | T6 | T7 | T8 | T9 | T10 | T11 | min | max | avg | abs. avg. |
|---|---|---|---|---|---|---|---|---|---|---|---|---|---|---|---|
| **HVR (our)** | 337.50 | 33.33 | 100.00 | 55.56 | 223.53 | 82.61 | 62.07 | 87.88 | 88.57 | 80.56 | 48.78 | 33.33 | 337.50 | 109.13 | 109.13 |
| HV | 75.00 | 33.33 | 100.00 | 83.33 | 158.82 | 21.74 | 58.62 | 39.39 | 57.14 | 41.67 | 70.73 | 21.74 | 158.82 | 67.25 | 67.25 |
| HR | -50.00 | 33.33 | 75.00 | 60.00 | 5.88 | 4.35 | 58.62 | 51.52 | 34.29 | 55.56 | 26.83 | -50.00 | 75.00 | 32.31 | 41.40 |
| VR | 0.00 | 33.33 | 25.00 | 126.67 | 23.53 | -17.39 | 24.14 | 0.00 | -5.71 | 27.78 | -17.07 | -17.39 | 126.67 | 20.02 | 27.33 |
| R | -25.00 | 33.33 | 37.50 | 26.67 | -17.65 | -13.04 | -24.14 | -27.27 | 5.71 | -27.78 | -17.07 | -27.78 | 37.50 | -4.43 | 23.20 |
| LLM | -25.00 | 33.33 | 37.50 | 6.67 | -41.18 | -21.74 | -24.14 | -6.06 | -8.57 | 8.33 | 2.44 | -41.18 | 37.50 | -3.49 | 19.54 |

In Table 15 we report the results for the metric Expanded Plan Verification (EPV) for HVR and all the baselines using Gemini as LLM-based planner. The results for each method are presented per task, along with the average across all 12 tasks.

*Table 15.* Results for the Expanded Plan Verification (EPV) metric using Gemini across all 12 tasks are presented as percentages, with task averages also provided.

| | Moderate complexity tasks | | | | | | High complexity tasks | | | | | | |
| --- | --- | --- | --- | --- | --- | --- | --- | --- | --- | --- | --- | --- | --- |
| | T1 | T2 | T3 | T4 | T5 | T6 | T7 | T8 | T9 | T10 | T11 | T12 | avg. |
| **HVR (our)** | 48.57 | 34.55 | 100.00 | 100.00 | 100.00 | 100.00 | 100.00 | 100.00 | 74.24 | 100.00 | 100.00 | 100.00 | 88.11 |
| HV | 100.00 | 100.00 | 100.00 | 100.00 | 100.00 | 100.00 | 100.00 | 100.00 | 100.00 | 100.00 | 100.00 | 100.00 | 100.00 |
| HR | 33.33 | 13.16 | 34.00 | 10.64 | 54.76 | 54.17 | 60.26 | 3.49 | 3.53 | 3.13 | 34.74 | 4.62 | 25.82 |
| VR | 42.11 | 51.22 | 48.89 | 12.82 | 53.85 | 50.00 | 13.24 | 13.04 | 14.08 | 35.90 | 10.59 | 40.00 | 32.14 |
| R | 29.41 | 20.59 | 6.52 | 12.20 | 8.11 | 54.17 | 12.96 | 5.00 | 12.00 | 3.85 | 4.62 | 14.89 | 15.36 |
| LLM | 29.41 | 10.00 | 6.82 | 12.20 | 8.82 | 54.17 | 12.96 | 4.48 | 4.29 | 8.14 | 5.13 | 40.00 | 16.37 |

In Table 16 we report the results for the metric Macro Plan Verification (MPV) for all the methods that use symbolic validation (i.e., HVR and HV) using Gemini as LLM-based planner. The results for each method are presented per task, along with the average across all 12 tasks. They are shown both before correction (first part of the table) and after correction (second part of the table).

*Table 16.* Results for the Macro Plan Verification (MPV) metric using Gemini across all 12 tasks are presented as percentages. The results are shown both before and after correction, with task averages also provided.

| | Moderate complexity tasks | | | | | | High complexity tasks | | | | | | |
| --- | --- | --- | --- | --- | --- | --- | --- | --- | --- | --- | --- | --- | --- |
| | T1 | T2 | T3 | T4 | T5 | T6 | T7 | T8 | T9 | T10 | T11 | T12 | avg. |
| | | | | | | Before correction | | | | | | | |
| **HVR (our)** | 100.00 | 100.00 | 100.00 | 100.00 | 100.00 | 100.00 | 100.00 | 100.00 | 100.00 | 100.00 | 100.00 | 100.00 | 100.00 |
| HV | 100.00 | 100.00 | 100.00 | 100.00 | 100.00 | 100.00 | 100.00 | 100.00 | 100.00 | 100.00 | 100.00 | 5.26 | 92.11 |
| | | | | | | After correction | | | | | | | |
| **HVR (our)** | 100.00 | 100.00 | 100.00 | 100.00 | 100.00 | 100.00 | 100.00 | 100.00 | 100.00 | 100.00 | 100.00 | 100.00 | 100.00 |
| HV | 100.00 | 100.00 | 100.00 | 100.00 | 100.00 | 100.00 | 100.00 | 100.00 | 100.00 | 100.00 | 100.00 | 100.00 | 100.00 |

In Table 17 we report the results for the metric Atomic Action Block Verification (AABV) for all the methods that use symbolic validation (i.e., HVR and HV) using Gemini as LLM-based planner. The results for each method are presented per task, along with the average across all 12 tasks. They are shown both before correction (first part of the table) and after correction (second part of the table). For comparison purposes, we also included results for HR in the first part of the table.

*Table 17.* Results for the Atomic Action Block Verification (AABV) metric using Gemini across all 12 tasks are presented as percentages. The results are shown both before and after correction, with task averages also provided.

| | Moderate complexity tasks | | | | | | High complexity tasks | | | | | | |
| --- | --- | --- | --- | --- | --- | --- | --- | --- | --- | --- | --- | --- | --- |
| | T1 | T2 | T3 | T4 | T5 | T6 | T7 | T8 | T9 | T10 | T11 | T12 | avg. |
| | | | | | | Before correction | | | | | | | |
| **HVR (our)** | 0.00 | 100.00 | 11.11 | 0.00 | 12.50 | 25.00 | 0.00 | 11.11 | 9.09 | 7.14 | 0.00 | 20.00 | 16.33 |
| HV | 50.00 | 100.00 | 12.50 | 28.57 | 0.00 | 33.33 | 100.00 | 100.00 | 9.09 | 30.00 | 31.25 | 10.53 | 42.11 |
| HR | 0.00 | 75.00 | 12.50 | 85.71 | 16.67 | 33.33 | 85.71 | 0.00 | 0.00 | 50.00 | 0.00 | 0.00 | 29.91 |
| | | | | | | After correction | | | | | | | |
| **HVR (our)** | 0.00 | 100.00 | 100.00 | 100.00 | 12.50 | 100.00 | 100.00 | 100.00 | 45.45 | 100.00 | 100.00 | 100.00 | 79.83 |
| HV | 100.00 | 100.00 | 100.00 | 100.00 | 100.00 | 100.00 | 100.00 | 100.00 | 100.00 | 100.00 | 100.00 | 100.00 | 100.00 |

