# OpenReview forum: "Hierarchical Planning for Complex Tasks with Knowledge Graph-RAG and Symbolic Verification"
_ICML.cc/2025/Conference — ICML 2025 poster_

### Official Review · Reviewer_Ac9Q · 2025-03-11

**Overall Recommendation:** 3

**Summary:**

This paper introduces HVR, a neuro-symbolic approach that enhances LLM-based planning by integrating hierarchical planning, retrieval-augmented generation (RAG) over knowledge graphs, and symbolic verification. The proposed method tackles long-horizon and complex task planning by decomposing tasks into macro actions and atomic actions, refining them through symbolic verification to ensure feasibility before execution. The Knowledge Graph-RAG (KG-RAG) component provides structured knowledge retrieval, improving accuracy while reducing hallucinations. The Symbolic Validator plays a dual role: first, it simulates plans in an "ideal world" for verification before execution; second, it functions as a failure detector, aligning expected world states with real-time observations. The system also builds a macro-action library, enabling knowledge transfer across agents by storing reusable action sequences. HVR is evaluated in AI2Thor, a kitchen-based robotic simulator, across 12 tasks of varying complexity, demonstrating significant performance improvements over baselines. Results show that RAG is crucial for smaller LLMs, while hierarchical planning is more impactful for larger models. Symbolic verification consistently enhances plan correctness, but LLMs still tend to generate unnecessarily long plans. The study highlights that LLM-based planners perform well on goal-oriented tasks but struggle with open-ended objectives, emphasizing the need for better failure detection and plan optimization techniques.

**Claims And Evidence:**

I think most of the claims are well supported, for example:
1. HVR improves task success rates through hierarchical planning, KG-RAG retrieval, and symbolic verification
- The paper evaluates HVR on AI2Thor, showing higher task success rates than baselines.
- The ablation study confirms that hierarchical planning, RAG, and symbolic verification each contribute to performance gains.
- Symbolic verification improves plan feasibility, and RAG reduces hallucinations, validating these components.
2. Symbolic verification consistently enhances plan correctness
- The Symbolic Validator verifies plans before execution, preventing physically impossible or illogical actions.
- Experiments show that removing the validator leads to lower success rates, confirming its importance.
- **Suggestion**: It would be better if the model compare with alternative approaches for consistency concern, e.g. LLM self-consistency, or self-checking before plan execution.

**Essential References Not Discussed:**

I think the discussion on related works is comprehensive.

**Experimental Designs Or Analyses:**

The selected benchmarks and metrics generally make sense to me. There are some suggestions I provide in the *Methods And Evaluation Criteria* section. Specifically:
1. AI2Thor is a widely used interactive simulation environment that allows for testing LLM-based planning agents in real-world-inspired tasks. It supports long-horizon goal-oriented tasks, which are crucial for evaluating structured planning. Prior work on embodied AI (e.g., ALFRED, BabyAI) has used similar environments, ensuring comparability.
2. The paper evaluates
- Task success rate – Measures whether the agent achieves the goal state.
- Plan efficiency – Measures action length and unnecessary steps.
These metrics are crucial to long-horizon planning, as it requires both correctness (success rate) and efficiency (fewer redundant steps).
Prior AI planning work (e.g., Hierarchical Task Networks (HTNs), PDDL-based planners) has used similar evaluation criteria.
3. The paper conducts ablation studies to test how much each component (hierarchical planning, KG-RAG, symbolic verification) contributes to performance. This isolates which modules are most crucial for performance improvements and helps clarify whether RAG and symbolic verification contribute independently or synergistically.

Suggestions:
1. Compare HVR to traditional symbolic planning systems (e.g., PDDL).
2. Evaluate plan robustness and **error recovery** to measure adaptability.

**Methods And Evaluation Criteria:**

Mostly yes, for example the idea about hierarchical planning and symbolic verification are well-suited for long-horizon task and are applicable in real-world planning problems. The selected benchmark and metrics like success rate and execution efficiency are also crucial in real-world applications.

Some suggestions:
1. The work mainly focuses on comparing with the variation of itself, e.g. HR, HV. But comparing with classical planning methods such as PDDL planners in the experiments is also necessary.
2. The renumbering task is good to show how the proposed method deals with specific operation order. It'd be better if you can also show and evaluate how the method can recover from errors. The current evaluation focuses on success rates, but some evaluation on how often the symbolic validator can detect errors and adjust the plan would be helpful.

**Other Comments Or Suggestions:**

N/A

**Other Strengths And Weaknesses:**

For weaknesses just the ones mentioned in previous sections.
For strengths, HVR presents a novel integration of hierarchical planning, structured knowledge retrieval (KG-RAG), and symbolic verification to improve long-horizon task execution using LLMs. This neuro-symbolic approach is an important step toward more structured and interpretable AI planning, moving beyond purely end-to-end LLM reasoning. The evaluation in AI2Thor demonstrates real-world applicability for embodied AI and assistive task planning, while the macro-action library enables knowledge transfer across tasks, making HVR significant for autonomous systems.

**Questions For Authors:**

No questions other than the ones in previous sections.

**Relation To Broader Scientific Literature:**

HVR builds upon prior work in hierarchical task planning, retrieval-augmented generation (RAG), symbolic reasoning, and LLM-based embodied AI. It extends classical AI planning methods by incorporating LLM-based reasoning with **symbolic verification** to ensure plan feasibility before execution. Compared to existing RAG-based AI agents like AutoGPT and ReAct, HVR leverages **structured knowledge retrieval (KG-RAG)** instead of unstructured document retrieval, reducing hallucinations in task planning. It also dynamically adjusts symbolic constraints using retrieved knowledge, enhancing planning accuracy in real-world tasks.

In the context of LLM-based task planning, HVR introduces a hierarchical macro-action library, similar to meta-learning frameworks and the options framework in reinforcement learning, to enable knowledge transfer across tasks. Compared to SayCan (Google DeepMind) and ALFRED (AI2Thor-based task planning models), HVR structures planning hierarchically and incorporates symbolic validation to improve long-horizon reasoning. However, the paper does not explicitly compare HVR to these models, making it unclear whether hierarchical planning, KG-RAG, and symbolic verification offer unique advantages over these prior methods. Including these comparisons would provide a clearer understanding of HVR’s contributions relative to existing AI planning systems.

**Theoretical Claims:**

The paper doesn't have formal mathematical proofs. The claims are mainly supported by evaluations. And I believe they are generally supported by those evidence.

---

> ### Author Rebuttal · Authors · 2025-03-31
>
> We thank the reviewer for their valuable feedback, which has greatly improved our paper. Below, we summarize the main concerns and detail the revisions and clarifications made to address them.
>
> **Q1: Missing comparison with the state-of-the-art**
>
> We did not include direct comparisons with existing methods, as none address complex, long-horizon kitchen tasks compatible with our setup. However, following reviewers’ suggestions, we considered additional works and included additional results/discussion in the Appendix of the revised paper, summarized below:
>
> * *SMART-LLM: Smart Multi-Agent Robot Task Planning using Large Language Models* This work is closely related to ours, as it uses the same simulator and addresses complex tasks (but in a multi-agent setting). We considered only 15 tasks, excluding those not related to the kitchen domain, of which only 12 were feasible in our setup due to a key design difference: unlike SMART-LLM, which assumes full knowledge of all object locations (including hidden ones), our system—aligned with RECOVER —only considers visible objects (hidden objects are treated as failures). HVR with Gemini-2.0-flash successfully planned and executed all 12 tasks.
> * *ProgPrompt: Generating Situated Robot Task Plans using Large Language Models* In this work there are 10 available tasks, 7 of which are kitchen related. We adapted the original implementation with the VirtualHome simulator to the AI2Thor simulator. Using Gemini-2.0-flash, HVR was able to plan and execute correctly all of the tasks. However, it's worth noting that these tasks are relatively simple compared to those in our benchmark. Additionally, ProgPrompt was evaluated using GPT-3, and their performance would likely improve with a more capable LLM.
> * *LLM+P: Empowering Large Language Models with Optimal Planning Proficiency* These methods are built on different simulators and/or involve tasks that are not compatible with our setup (e.g., organizing blocks on a table). As a result, a direct comparison would require a substantial and non-trivial reimplementation, which falls outside the scope of this work.
> * *PDDL-based planning systems* We could not compare with PDDL-based planning systems on our tasks as we found no existing works using planners expressive enough. As described in Section 3.3, our system uses conjunctive, disjunctive, and conditional PDDL statements. We therefore implemented a custom validator in Python, adapted specifically to the AI2Thor environment.
> * *ISR-LLM: Iterative Self-Refined Large Language Model for Long-Horizon Sequential Task Planning* Here, given specifications of simple abstract tasks in a simulated kitchen environment, the authors generate both a PDDL domain and a goal state. While this can work for a simple environment with few available actions, we argue it is generally not possible to generate PDDL domain specifications from high-level natural language task specifications such as those we use in our work. Thus, we did not run experimental comparisons with this method.
> * *Translating Natural Language to Planning Goals with Large-Language Models* This work uses the ALFRED simulator, similar to AI2Thor. However, the implementation is too limited for our use case and does not support key functionalities like creating individual slices after slicing an object, or modeling interactions such as opening appliances—essential for our tasks— making it unsuitable for a meaningful comparison.
>
> **Q2: Evaluation of Error Detection/Recovery and System Robustness**
>
> We thank the reviewer for the suggestion. The metrics EPV, MPV, and AABV (highlighted as blue metrics in the ‘Verification’ part of  in Table 2) are specifically designed to evaluate the role of symbolic verification and correction. These metrics are defined as follows:
> (4) Expanded Plan Verification (EPV) indicates the extent to which the expanded plan (full sequence of atomic ac- tions) has been successfully verified. It is calculated as the ratio of verified steps to the total number of steps in the generated plan. (5) Macro Plan Verification (MPV) measures the extent to which the macro plan has been verified. It is calculated as the ratio of verified macro plan steps to the total number of macro steps. (6) Atomic Action Block Verification (AABV) evaluates the extent to which the macro plan has been verified at the level of atomic action blocks. It is determined by dividing the number of verified atomic action blocks by the total number of macro actions.
>
> To summarize, these metrics reflect how effectively the symbolic validator detects and helps recover from errors at different levels of the plan (expanded, macro, and atomic action blocks). Their strong correlation with Plan Correctness (PC) demonstrates that symbolic validation not only identifies errors but also leads to measurable improvements in plan quality. This provides insight into the system’s robustness and adaptability in recovering from planning errors.

---

> > ### Comment · Reviewer_Ac9Q · 2025-04-03
> >
> > Thanks for your response! The rebuttal about Q1 does make the work more complete in my view. As I read through other reviewers comment, I'd keep my current rating.

---

### Official Review · Reviewer_9MAP · 2025-03-11

**Overall Recommendation:** 3

**Summary:**

The authors propose a neuro-symbolic approach that combines LLMs-based planners with Knowledge Graph-based RAG for hierarchical plan generation. It breaks down complex tasks into subtasks and then into executable atomic action sequences. A symbolic validator is integrated to ensure formal correctness, task decomposition, and to detect failures by comparing expected and observed world states.

## update after rebuttal
I will maintain my positive opinion.

**Claims And Evidence:**

Yes.

**Essential References Not Discussed:**

No.

**Experimental Designs Or Analyses:**

Yes.

**Methods And Evaluation Criteria:**

Yes.

**Other Comments Or Suggestions:**

The comparison of efficiency and ablation experiments should be further improved.

**Other Strengths And Weaknesses:**

Strengths:
1. The proposed framework is reasonable. It provides a new idea for  LLM planning through a novel neuro-symbolic integration.

2. HVR shows obvious improvements in performance.

3. The result analysis is in-depth, strongly supporting the viewpoints of the paper.

Weaknesses:

The issue of efficiency has not been discussed, nor has it been compared in the experimental section. Conducting planning and retrieval augmentation at two granularities, as well as failure detection, constitutes a relatively complex process. It is unacceptable that there is no comparison of efficiency.

**Questions For Authors:**

No questions.

**Relation To Broader Scientific Literature:**

HVR enhances LLM planning capabilities through a novel neuro-symbolic integration of Hierarchical planning, symbolic Verification and reasoning, and RAG methods over symbolic Knowledge Graphs.

**Theoretical Claims:**

No theoretical claims.

---

> ### Author Rebuttal · Authors · 2025-03-31
>
> We thank the reviewer for their valuable feedback, which has greatly improved our paper. Below, we summarize the main concerns and detail the revisions and clarifications made to address them.
>
> **Q1: Missing comparison and discussion of efficiency**
>
> We have included a study of the efficiency of our method including 2 new plots in a new section in the appendix "Efficiency considerations" in the revised paper:
>
> Figure A (*Average execution times (in seconds) per model using Gemini 1.5 flash*) shows the average computational time for the different models across the 13 tasks using Gemini-1.5-flash as LLM. As expected, approaches involving hierarchical decomposition  require (approximately three times) more time than those without.
> However, as LLMs become faster, the overhead introduced by the HVR framework becomes increasingly negligible, while substantially improving plan correctness.
> For example, running the same tasks with Gemini-2.0-flash reduced the average execution time for HVR from 3285.32 seconds to 681.51 seconds—a 5x improvement.
>
> In contrast to this improvement, relying solely on an LLM as a planner demands a substantially larger context window, which quickly becomes impractical as task or environment complexity increases. HVR addresses this limitation through retrieval-augmented generation (RAG), enabling it to dynamically access only the relevant information from the knowledge graph. This design ensures stable processing times and scalability, even as the environment grows, whereas LLM-only approaches face escalating computational demands and eventual context window exhaustion.
>
> Figure B (*Plan Correctness vs. Execution Time in seconds with Gemini 1.5 flash*) shows the trade-off between plan correctness and execution time across the different methods.
> While HVR takes longer to compute plans, it consistently achieves the highest correctness. In contrast, simpler methods like LLM or R achieve faster execution but produce significantly less reliable plans, demonstrating the benefit of HVR’s more structured approach.

---

### Official Review · Reviewer_ojF6 · 2025-03-13

**Overall Recommendation:** 1

**Summary:**

This paper introduces HVR, a task planning method that integrates hierarchical planning, retrieval-augmented generation (RAG) over symbolic knowledge graphs, and formal verification to enhance the performance of large language models (LLMs) in complex task planning. The proposed method decomposes the language-described tasks into manageable macro actions and further into atomic actions, ensuring formal correctness through symbolic validation. Experimental results in the AI2Thor kitchen environment demonstrate that HVR outperforms baseline methods.

## update after rebuttal
In settings where a precise world model exists, the world state is fully known, and the problem is guaranteed to be solvable, symbolic planners are already highly effective at solving planning problems. Therefore, a more meaningful and impactful research direction is to explore the use of LLMs for planning in scenarios where the world model is incomplete or unavailable. Extending HVR to operate under such conditions would significantly enhance its practical value and increase its potential for broader acceptance.

**Claims And Evidence:**

In this paper, the authors assume that a precise domain model is known (as stated in lines 194-198 on page 4) and that the environment state can be obtained through OntoThor. Given these assumptions, a classical planner (e.g., Fast Downward) should be able to generate a valid plan with the PDDL-style goal specification. Why is it necessary to use an LLM for planning in this setting? It seems that the LLM’s role could be limited to converting natural language task descriptions into PDDL goal specifications, as done in [1] and [2], after which classical planning would likely achieve a high success rate. The key advantage of using an LLM for planning is its ability to operate without a precisely defined domain model, which can be complex to construct. However, this paper still relies on an exact domain model, which raises questions about the necessity of employing an LLM for planning in this context.

References:
[1] Translating Natural Language to Planning Goals with Large-Language Models, ARXIV 2023.
[2] LLM+P: Empowering Large Language Models with Optimal Planning Proficiency, ARXIV 2023.

**Essential References Not Discussed:**

See [1] and [2] mentioned in Claims And Evidence.

**Experimental Designs Or Analyses:**

Yes. The design of the baselines serves as an ablation study of the complete method, allowing the effectiveness of the proposed components to be validated through experiments.

However, it lacks a comparison with code-style task planning approaches, such as ProgPrompt [3].

Reference:
[3] ProgPrompt: Generating Situated Robot Task Plans using Large Language Models, ICRA 2023.

**Methods And Evaluation Criteria:**

Yes.

**Other Comments Or Suggestions:**

No.

**Other Strengths And Weaknesses:**

Strengths:
1. The proposed method achieves good performance on various tasks in the AI2THOR environment.
2. Several new metrics are valuable for evaluating the methods.

Weaknesses:
1. The paper lacks a discussion about the limitations of the proposed method.

**Questions For Authors:**

See the comments under Claims and Evidence, Experimental Designs or Analyses, and Weaknesses.

**Relation To Broader Scientific Literature:**

Hierarchical planning helps prevent oversimplification or the omission of essential steps in long-horizon tasks.

**Theoretical Claims:**

No, there is no theoretical claim.

---

> ### Author Rebuttal · Authors · 2025-03-31
>
> We thank the reviewer for their valuable feedback, which has greatly improved our paper. Below, we summarize the main concerns and detail the revisions made to address them.
>
> **Q1: Why is it necessary to use an LLM for planning in this setting?**
>
> While prior works such as [1] and [2] show how natural language (NL) instructions can be translated into PDDL goals, they are limited to simple tasks and lack the flexibility needed for more complex scenarios. Moreover, symbolic planners tend to be brittle—minor changes in the environment or execution failures can break the entire plan. In contrast, HVR leverages the flexibility of LLMs throughout the whole planning process, addressing several key limitations of symbolic–only methods:
> * Partial plan execution: for particularly complex tasks, LLM-based models are able to produce at least some correct initial steps or an initial sub-task, while symbolic planners can only generate full plans and thus likely failing entirely on these type of tasks
> * LLMs enable dynamic adaptation during task execution, incorporating feedback from the environment to replan on the fly. Symbolic planners would need new mid-task NL instructions.
> * Thanks to LLMs, and even more thanks to the employment of retrieval-augmented generation (RAG), our approach achieves significantly faster planning, avoiding the exponential search space growth typical of classical planners.
>
> Lastly, although HVR currently relies on a fixed world model, it can be extended to operate without one—just as we generate macro-actions and their pre/post-conditions dynamically.
>
> **Q2: Missing comparison with the state-of-the-art**
>
> We did not include direct comparisons with existing methods, as none address complex, long-horizon kitchen tasks compatible with our setup. However, following reviewers’ suggestions, we considered additional works and included new results/discussions in the appendix of the revised paper. See the full response to *Review-Ac9Q*, summarized here:
> * *SMART-LLM: Smart Multi-Agent Robot Task Planning using Large Language Models* This work is closely related to ours, as it uses the same simulator and addresses complex tasks (but in a multi-agent setting). HVR with Gemini-2.0-flash successfully planned and executed all 12 tasks.
> * *ProgPrompt: Generating Situated Robot Task Plans using Large Language Models* We adapted the original implementation with the VirtualHome simulator to the AI2Thor simulator. HVR with Gemini-2.0-flash was able to plan and execute correctly all kitchen-related tasks.
> * *LLM+P: Empowering Large Language Models with Optimal Planning Proficiency* These methods are built on different simulators and/or involve tasks that are not compatible with our setup.
> * *PDDL-based planning systems* We could not compare with PDDL-based planning systems on our tasks as we found no existing works using planners expressive enough.
> * *Translating Natural Language to Planning Goals with Large-Language Models* This work uses the ALFRED simulator, similar to AI2Thor. However, the implementation is too limited for our use case and does not support key functionalities making it unsuitable for a meaningful comparison.
>
> **Q3: Missing discussion about HVR limitations**
>
> We have included a discussion of our work’s limitations and future work in a new section in the appendix in the revised paper:
>
> The HVR framework integrates hierarchical planning, knowledge graph retrieval, and symbolic validation to enhance LLM-based task planning, but it also presents several limitations that point to promising directions for future work.
> One key limitation is its reliance on a fixed ontology and action space, which constrains generalization to new environments or tasks.  While updating to the ontology currently requires expert domain knowledge, the predefined action space could be extended to include novel actions automatically using LLM-based techniques—similar to how HVR currently generates macro actions along with their corresponding pre- and post-conditions.
> Prompt sensitivity is another concern common to LLM-based systems, although recent models show improved robustness to variations in prompt phrasing. Additionally, HVR's current use of natural language to mediate interactions between the LLM and the knowledge graph may not be the most efficient; leveraging sub-symbolic or embedded representations could improve this integration.
> Another limitation is the current restriction to a linear structure for the generated plans. Supporting partial-order plans would not only allow for more flexible planning but also make it possible to execute different branches in parallel, which is especially valuable in multi-agent settings.
> Finally, in our method, error correction is done independently for each part of the plan and does not address interdependencies between errors at different planning levels. A more connected correction strategy that reasons across macro and atomic actions could help improve overall correctness.

---

> > ### Comment · Reviewer_ojF6 · 2025-04-08
> >
> > Thank you for your response. I fully understand that traditional symbolic planning methods are inherently brittle, as they rely heavily on an accurate and complete world model to produce valid plans. However, under the setting assumed in this paper—where such a precise world model exists, the world state is known, and the problem is guaranteed to be solvable—symbolic planners are already capable of solving the planning problem effectively. Therefore, in this particular setting, I believe the use of LLMs for planning lacks clear motivation or necessity. Instead, the more meaningful and valuable direction is to explore the use of LLMs for planning when such a precise world model is unavailable or incomplete.
> >
> > As the authors have mentioned in the rebuttal, HVR can be extended to operate without a precise world model. I believe that once such an extension is realized, the approach would be much more valuable and well-suited for acceptance.
> >
> > In addition, regarding the missing comparison with methods [1] and [2], it is worth noting that they mainly utilize large language models for translation purposes. Given the availability of the world model and world state in this work, along with the open-source Fast Downward planner, it should not be difficult to adapt these two approaches to the tasks presented in this paper, based on my experience. Therefore, I believe future extensions of this work should include these approaches in the experimental comparison.

---

### Official Review · Reviewer_HvPs · 2025-03-14

**Overall Recommendation:** 3

**Summary:**

This paper proposes a LLM-based approach (HAR) to tackle long-horizon and complex robotic planning, which integrates hierarchical planning and Retrieval-Augmented Generation (RAG). Specifically, HAR leverages the LLM to decompose complex tasks into subtasks at different abstraction levels while integrates the RAG method to retrieve relevant context from the agent’s knowledge graph for plan generation. Then, HAR employs a Symbolic Validator to verify and correct t the generated plans. Experiments on multiple datasets of varying difficulty levels demonstrate the effectiveness of the HAR method across multiple LLMs.

**Claims And Evidence:**

Most claims are supported. However, in Section 2.5, the authors mention that one advantage of this method is its ability to build a reusable library of macro actions, but they do not provide experimental evidence to support its effectiveness.

**Essential References Not Discussed:**

S. S. Kannan, V. L. N. Venkatesh and B. -C. Min, "SMART-LLM: Smart Multi-Agent Robot Task Planning using Large Language Models," 2024 IEEE/RSJ International Conference on Intelligent Robots and Systems (IROS)

Z. Zhou, J. Song, K. Yao, Z. Shu and L. Ma, "ISR-LLM: Iterative Self-Refined Large Language Model for Long-Horizon Sequential Task Planning," 2024 IEEE International Conference on Robotics and Automation (ICRA)

**Experimental Designs Or Analyses:**

Yes. This paper designs comprehensive and well-reasoned evaluation metrics to thoroughly assess the effectiveness of the proposed method. Additionally, through extensive ablation experiments, it validates the effectiveness of each component.

**Methods And Evaluation Criteria:**

Yes

**Other Comments Or Suggestions:**

Please refer to Other Strengths And Weaknesses

**Other Strengths And Weaknesses:**

Strengths：
1. A novel planning framework that integrates hierarchical planning, RAG, and symbolic validation.
2. Comprehensive and clear ablation experiments validate the effectiveness of each component.
3. Designed effective and sufficient experimental evaluation metrics.

Weaknesses：
1. The sufficiency of the experiments has certain limitations. The paper conducts experiments on only one dataset and does not include comparative experiments with existing methods（smart-llm）.

**Questions For Authors:**

Please refer to Other Strengths And Weaknesses

**Relation To Broader Scientific Literature:**

This paper leverages the advantages of existing methods to propose a novel methodological framework.

**Theoretical Claims:**

No theoretical proof.

---

> ### Author Rebuttal · Authors · 2025-03-31
>
> We thank the reviewer for their valuable feedback, which has greatly improved our paper. Below, we summarize the main concerns and detail the revisions and clarifications made to address them.
>
> **Q1: Missing details and experiments regarding the reusable library of macro actions**
>
> The macro actions are stored in the ontology following the same approach used in Cornelio & Diab 2024. While we have not yet conducted experiments leveraging the macro action library to accelerate the planning process, we plan to explore this in the future. We added additional details in a new section in the appendix, summarized below:
>
> OntoThor contains the class Action representing agent-environment interactions. We refine this by introducing two subclasses: Atomic Action (AA), as the original Action class, and Macro Action (MA) representing higher-level operations like boil-water.
> During execution, AA and MA instances are stored in the agent’s Knowledge Graph. Each MA instance is linked to its NL description including pre- and post-conditions, via the hasDescription predicate. These conditions can also be modeled as triples, extending the scene-graph representation in OntoThor.
> Each MA instance is also linked to its sequence of AAs via the hasAtomicAction predicate. After task execution, the set of MAs, along with their associated conditions, are stored in the Knowledge Graph and, if validated by the environment (see section 2.4), also added to the ontology.
>
> **Q2: Missing References**
>
> Thank you for the suggested references. We’ve added the missing citations and included SMART-LLM—previously unknown to us—in the revised manuscript, along with an experimental comparison (see Q3).
>
> **Q3: Missing comparison with State-of-the-art systems**
>
> We did not include direct comparisons with existing methods, as none address complex, long-horizon kitchen tasks compatible with our setup. However, following reviewers’ suggestions, we considered additional works and included additional results/discussion in the Appendix of the revised paper, summarized below:
> * *SMART-LLM: Smart Multi-Agent Robot Task Planning using Large Language Models*
> This work is closely related to ours, as it uses the same simulator and addresses complex tasks (but in a multi-agent setting). We considered only 15 tasks, excluding those not related to the kitchen domain, of which only 12 were feasible in our setup due to a key design difference: unlike SMART-LLM, which assumes full knowledge of all object locations (including hidden ones), our system—aligned with RECOVER —only considers visible objects (hidden objects are treated as failures). HVR with Gemini-2.0-flash successfully planned and executed all 12 tasks.
> * *ProgPrompt: Generating Situated Robot Task Plans using Large Language Models*
> In this work there are 10 available tasks, 7 of which are kitchen related. We adapted the original implementation with the VirtualHome simulator to the AI2Thor simulator. Using Gemini-2.0-flash, HVR was able to plan and execute correctly all of the tasks. However, it's worth noting that these tasks are relatively simple compared to those in our benchmark. Additionally, ProgPrompt was evaluated using GPT-3, and their performance would likely improve with a more capable LLM.
> * *LLM+P: Empowering Large Language Models with Optimal Planning Proficiency*
> These methods are built on different simulators and/or involve tasks that are not compatible with our setup (e.g., organizing blocks on a table). As a result, a direct comparison would require a substantial and non-trivial reimplementation, which falls outside the scope of this work.
> * *PDDL-based planning systems*
> We could not compare with PDDL-based planning systems on our tasks as we found no existing works using planners expressive enough. As described in Section 3.3, our system uses conjunctive, disjunctive, and conditional PDDL statements. We therefore implemented a custom validator in Python, adapted specifically to the AI2Thor environment.
> * *ISR-LLM: Iterative Self-Refined Large Language Model for Long-Horizon Sequential Task Planning*
> Here, given specifications of simple abstract tasks in a simulated kitchen environment, the authors generate both a PDDL domain and a goal state. While this can work for a simple environment with few available actions, we argue it is generally not possible to generate PDDL domain specifications from high-level natural language task specifications such as those we use in our work. Thus, we did not run experimental comparisons with this method.
> * *Translating Natural Language to Planning Goals with Large-Language Models*
> This work uses the ALFRED simulator, similar to AI2Thor. However, the implementation is too limited for our use case and does not support key functionalities like creating individual slices after slicing an object, or modeling interactions such as opening appliances—essential for our tasks— making it unsuitable for a meaningful comparison.

---

### Decision · Program_Chairs · 2025-05-01

**Decision:**

Accept (poster)

**Comment:**

This paper received four high-quality reviews (scores: 3, 3, 3, 1).

The three positive reviewers appreciated the extensive experiments and the innovative methodology, which effectively integrates hierarchical planning, knowledge graph-based retrieval augmentation (KG-RAG), and symbolic verification within a unified neuro-symbolic framework. The reviewers highlighted the paper's strong empirical results, clearly demonstrating improved task success rates and robustness compared to baseline methods.

The negative reviewer raised concerns regarding the necessity of employing large language models (LLMs) in scenarios where symbolic planners are traditionally sufficient due to precise domain models. In the rebuttal, the authors convincingly addressed this concern by clarifying the unique advantages of their approach. The authors emphasized that their use of LLMs enhances planning flexibility, supports partial plan execution, and allows dynamic adaptation during execution, effectively overcoming limitations inherent to purely symbolic methods.

Having thoroughly reviewed the paper, carefully considered all reviewer comments, and evaluated the authors' detailed rebuttal, the AC concurs with the positive reviewers. The innovative integration proposed, along with robust experimental validation, clearly justifies a clear acceptance.